

# Measurement induced transitions in non-Markovian free fermion ladders

Mikheil Tsitsishvili[1,2]⋆, Dario Poletti[1,3,4,5],
Marcello Dalmonte[1,2] and Giuliano Chiriacò[1,2,6,7]†

**1** The Abdus Salam International Centre for Theoretical Physics (ICTP), Trieste, Italy
**2** SISSA — International School of Advanced Studies, Trieste, Italy
**3** Science, Mathematics and Technology Cluster,
Singapore University of Technology and Design, Singapore
**4** Engineering Product Development Pillar,
Singapore University of Technology and Design, Singapore
**5** Centre for Quantum Technologies, National University of Singapore 117543, Singapore
**6** Dipartimento di Fisica e Astronomia "Ettore Majorana", Università di Catania, Catania, Italy
**7** INFN, Sez. Catania, I-95123 Catania, Italy

⋆ mtsitsis@ictp.it , † giuliano.chiriaco@dfa.unict.it

## Abstract

Recently there has been an intense effort to understand measurement induced transitions, but we still lack a good understanding of non-Markovian effects on these phenomena. To that end, we consider two coupled chains of free fermions, one acting as the system of interest, and one as a bath. The bath chain is subject to Markovian measurements, resulting in an effective non-Markovian dissipative dynamics acting on the system chain which is still amenable to numerical studies in terms of quantum trajectories. Within this setting, we study the entanglement within the system chain, and use it to characterize the phase diagram depending on the ladder hopping parameters and on the measurement probability. For the case of pure state evolution, the system is in an area law phase when the internal hopping of the bath chain is small, while a non-area law phase appears when the dynamics of the bath is fast. The non-area law exhibits a logarithmic scaling of the entropy compatible with a conformal phase, but also displays linear corrections for the finite system sizes we can study. For the case of mixed state evolution, we instead observe regions with both area, and non-area scaling of the entanglement negativity. We quantify the non-Markovianity of the system chain dynamics and find that for the regimes of parameters we study, a stronger non-Markovianity is associated to a larger entanglement within the system.

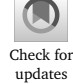

# 1 Introduction

In most contexts of quantum physics and quantum technologies, systems are subject to unitary evolution plus a certain degree of dissipative dynamics [1,2]. The latter arises from interaction with the environment or with an external observer, and is a fundamental characteristic of realistic models of quantum systems. The interplay between unitary dynamics and dissipation has been widely studied in the last years and has been shown to induce a great variety of phenomena in solid state, cold atoms or quantum computing systems. For example, interaction with an external drive may induce dissipative phase transitions, dynamical transitions into metastable or non-equilibrium steady states, transitions of the entanglement, slow relaxation dynamics, emergence of time crystals, etc. [3–33].

In particular, measurement induced entanglement transitions [26–28, 34–60] have been the subject of intense research in recent years. They appear at the level of the scaling behavior of the entanglement with the size of the system, and are caused by the action of random measurements that collapse the entanglement of the system, and counteract the correlation spreading action of the unitary dynamics. Even at a finite rate of measurement, this interplay leads to a transition between phases with extensive (or critical) scaling of the entanglement, and phases with low entanglement. Indeed, entanglement transitions have been observed in many different settings and models, including random circuits [27, 28, 36, 61–63], stabilizer and Clifford circuits [44, 45, 64–69], Ising-like models with either short-range or long-range interactions [53, 70–72], and systems of free fermions [56–58, 73–76]. In all these works, the studied systems are coupled to Markovian environments, which cause memory-less measurements – i.e. the probability of a random measurement occurring is completely independent of the history of the system.

In a recent work [77] that we co-authored, we analyzed the entanglement transition in system with a non-Markovian dissipative processes. We investigated the entanglement in random unitary circuits by unravelling the many-body dynamics and calculating analytically the effect of non-Markovianity on the probability of dissipative measurements. This study is a step

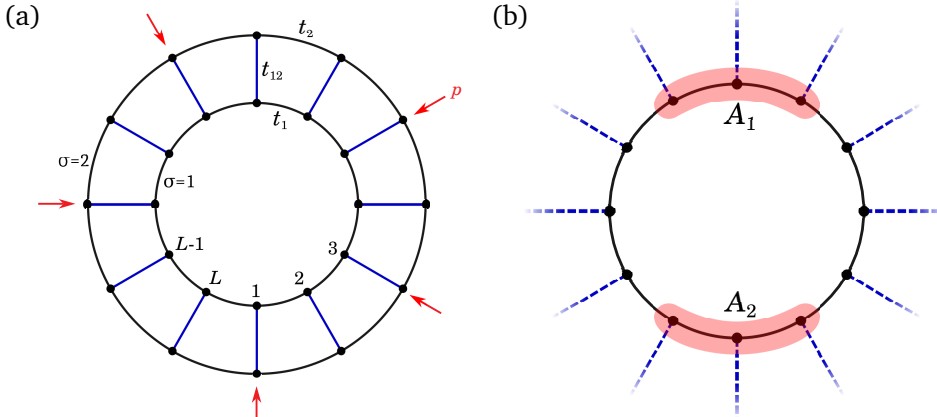

Figure 1: (a) A pictorial representation of a fermionic ladder with the periodic boundary conditions. The tunneling amplitude within the outer ($\sigma = 2$) and the inner chain ($\sigma = 1$) is $t_2$ and $t_1$, respectively. $t_{12}$ is the inter-chain tunneling amplitude. The red arrows indicate the temporally and spatially random projective measurements of the particle occupation at the corresponding sites. $p$ is the probability of performing such measurement. (b) After integrating out the degrees of freedom of the outer chain, the inner chain is partitioned into segments $\{A_j\}$. The residual correlations between outer and inner chain is pictorially represented by dashed blue lines. The size and location of $\{A_j\}$ segments can be arbitrary. Here we show a tri-partition of the inner chain into segments $A_1$, $A_2$ and the rest of the chain, labeled as $B$ throughout the paper.

forward in the field of transition induced by external baths, since most environments display memory effects and thus can be better described by a non-Markovian dynamics.

In the present work we study the non-Markovian entanglement transition using a fundamentally different approach which can be better extended to many-body interacting systems. We reproduce a non-Markovian dynamics by considering two coupled fermionic chains: the first one is the system under cosideration, while the second one acts as a bath with a nontrivial dynamics. The bath chain is also subject to a Markovian dissipative dynamics; thanks to the bath internal dynamics, which introduces memory effects, the system chain is effectively subject to a non-Markovian dissipation. By including explicitly the bath in our analysis, we pay a price in doubling the degrees of freedom, but are able to numerically simulate the quantum trajectories of the dynamics of the system and bath chain, which is purely Markovian. This approach is akin to the techniques exploiting an auxiliary extension of the Hilbert space of the system [78–81], which are also used in quantum thermodynamics under the name of super-bath approach [82–84].

More specifically, we consider a model of free fermions with next neighbors hopping within each chain and inter-chain hopping, see Fig.(1a). We assume the dissipative dynamics on the bath chain to be given by local projective measurements of the particle number. Both unitary and dissipative dynamics preserve the Gaussianity of the state of the system, thus allowing to express all the relevant observables in terms of two-point correlation functions and to perform efficient numerical simulations up to system sizes of hundreds of sites [56–58, 85–88].

We study the evolution of both chains until a steady state is reached, and then integrate out the bath chain, resulting in an effective non-Markovian dynamics for the remaining chain, see Fig.(1b). We then consider suitable partitions of the system chain and compute the average over the quantum trajectories of entanglement witnessing operators, such as entropy, mutual information [28, 57, 89] and negativity [90, 91].

The entanglement transition in Markovian free fermion systems has been recently investigated in several works, and has sparked a debate on the nature of the entangled phase in the low measurement regime and on the survival of the transition in the thermodynamic limit [57,58]. From a different perspective, our work helps shed light on the nature of the entanglement transition in free fermion systems.

For example, we find that even in the extreme regime in which the bath chain is always subject to measurements, the non-area-law phase still survives in the system chain. This is due to the coupling and internal hopping structure of the two chains: for example the coupling between the two chains can be weak enough that the system chain does not feel the measurements on the baths; or the hopping in the bath is good enough at scrambling information after measurements, so that entanglement loss in the system is very small.

We study the size scaling of entanglement entropy in the non-area-law phase, and find that it exhibits a logarithmic scaling compatible with a conformal field theory phase. At larger values of the bath chain hopping we observe linear corrections, which are likely due to a finite size effect and not an indicator of a volume law phase. In fact, an analysis of the mutual information indicates the presence of long-range correlations for all values of the bath chain hopping, which is not compatible with a volume law phase.

We also study the regime where the measurement probability is smaller than one. This case is fundamentally different, as the system chain is always in a mixed state. Thus the entropy and the mutual information are no longer good observables to study entanglement, as they also track classical correlations. We thus employ the fermionic negativity [88,92,93] to perform an analysis of the phase transition as function of the measurement probability. We find that the area law disappears for sufficiently weak measurements, and that the non-area-law phase exhibits a scaling behavior qualitatively similar to that of the entanglement entropy, with a mixture of linear and logarithmic contributions. Since in this case the negativity is the only observable at our disposal, we cannot attribute in a definitive way its scaling behavior to one scaling or the other (even if, in certain regimes, a volume scaling appears considerably more likely).

We then study how much the dynamics of the system is non-Markovian using already tested non-Markovianity measures [94–105]. This analysis is quite complex to perform as we need to use exact diagonalization techniques and simulate the dynamics many times, and thus the maximum system sizes that we can consider are limited. We find that the dynamics is non-Markovian in all the regions of the phase diagram as a function of the system and bath parameters, and that the degree of non-Markovianity changes and displays a pattern similar to that of the entanglement phase. In particular, we observe that a stronger degree of non-Markovianity is associated to a larger entanglement within the system.

The rest of the paper is organized as follows. In Section 2 we present the model and describe its features. In Section 3 we show our numerical results on the entanglement transition and analyze the phase diagram of the system. In Section 4 we discuss and quantify the non-Markovianity of the system dynamics. Finally in Section 5 we draw our conclusions.

## 2 The model

Our goal is to study a model whose partitioning into a bath component and a system component can give us insights into the non-Markovian dynamics of the system component. We focus on a model of two coupled chains of free and spinless fermions, with an approach that may be reminiscent of those based on doubling the Hilbert space for non-Markovian systems [78–81]. We consider periodic boundary conditions so that the geometry is that of a circular ladder. The legs of the ladder are the intrachain hoppings, while the rungs represent the interchain

coupling, as shown in Fig.(1a). The outer chain is the bath, while the inner chain is the system under study.

The ladder is at half filling[1] and evolves with a stroboscopic dynamics of time period $\tau_u$. Each cycle is constituted by a unitary evolution that lasts for the entire period $\tau_u$ and is governed by the Hamiltonian $\hat{H}$, and by projective measurements of the particle occupation on the outer chain, that occur at the end of the cycle. This simulates a non-Markovian dissipation when the degrees of freedom of the outer chain are traced out, pictorially represented on Fig.(1b).

The model Hamiltonian governing the unitary part of the evolution during time $\tau_u$ is

$$\hat{H} = \sum_{i,\sigma} t_\sigma \hat{c}^\dagger_{i,\sigma} \hat{c}_{i+1,\sigma} + t_{12} \sum_{i=1}^{L} \hat{c}^\dagger_{i,1} \hat{c}_{i,2} + \text{h.c.}, \tag{1}$$

where $\hat{c}^\dagger_{j,\sigma}, \hat{c}_{j,\sigma}$ are the fermionic creation and destruction operators on site $j$ of chain $\sigma = 1, 2$. We impose periodic boundary conditions as $\hat{c}_{L+n,\sigma} = \hat{c}_{n,\sigma}$.

The Hamiltonian can be diagonalized in Fourier space, where it is written as

$$\hat{H} = \sum_k \hat{\psi}^\dagger_k H_k \hat{\psi}_k, \qquad H_k = \begin{pmatrix} 2t_1 \cos k & t_{12} \\ t_{12} & 2t_2 \cos k \end{pmatrix}, \tag{2}$$

with $\hat{\psi}_k \equiv \begin{pmatrix} \hat{c}_{k,1} \\ \hat{c}_{k,2} \end{pmatrix}$ and $\hat{c}_{k,\sigma} = \sum_j e^{-ijk} \hat{c}_{j,\sigma} / \sqrt{L}$.

The unitary evolution operator over one cycle factorizes as $\hat{U} = \bigotimes_k \hat{U}_k$, where $\hat{U}_k = e^{-i\tau_u H_k}$ is the evolution operator on the subspace with momentum $k$:

$$\hat{U}_k = e^{-it\cos k \tau_u} \left[ \cos\left(\sqrt{t_{12}^2 + \delta^2 \cos^2 k}\, \tau_u\right) - i \frac{t_{12}\sigma^x + \delta \cos k \sigma^z}{\sqrt{t_{12}^2 + \delta^2 \cos^2 k}} \sin\left(\sqrt{t_{12}^2 + \delta^2 \cos^2 k}\, \tau_u\right) \right], \tag{3}$$

where $t = t_1 + t_2$ and $\delta = t_1 - t_2$. The identity and the Pauli matrices $\sigma^{x,z}$ operate in the chain index space. We work in the units of $\hbar = 1$ and fix $t_1 = 1$.

From Eq.(3) we recognize the periodic structure of $\hat{U}_k$ in $t_{12}$. In particular, we see that only the term proportional to $\sigma^x$ couples the two chains. Therefore when $t_{12} = 0$ the two chains are decoupled, as expected. This also occurs when $\sin\left(\sqrt{t_{12}^2 + \delta^2 \cos^2 k}\, \tau_u\right) = 0$, which in general cannot be satisfied at once for all values of $k$. However, when $\delta = 0$ (i.e. when the intrachain hoppings are equal), the decoupling condition reduces to $t_{12}\tau_u = n\pi$ showing the periodicity in $t_{12}$ of the dynamics on the special line $t_2 = t_1$. For small values of $\delta$ around this special line, the two chains are "quasi-decoupled" for $t_{12}\tau_u \approx n\pi, n \in \mathbb{Z}$ in the sense that the off-diagonal matrix elements in $\hat{U}_k$ are very small for every $k$.

Since the Hamiltonian is quadratic, the Gaussianity of the state is intact during the unitary evolution and Wick's theorem is applicable. After a time $\tau_u$, the unitary evolution of the state $|\Psi(t)\rangle \to |\Psi(t + \tau_u)\rangle = \hat{U}|\Psi(t)\rangle$ is interrupted and the system interacts with a local measuring apparatus. The random local measurements occur within a very short time, so that they may be considered instantaneous. The local particle occupation number on the outer chain, i.e. $\hat{n}_{i,2} = \hat{c}^\dagger_{i,2}\hat{c}_{i,2}$, is randomly measured with probability $p$ for each site of the outer chain.

In general, the projective measurements can spoil the Gaussianity of the state, however the type of the measurements that we consider does not [58]. This allows us to extract all

---

[1] Note that the total number of fermions in the ladder is conserved, but the number of fermions in each chain can change during the evolution.

relevant information regarding the state of the system from the two-point correlation matrix $\mathcal{D}_{ij,\sigma\sigma'}(t) = \langle\Psi(t)|\hat{c}^\dagger_{i,\sigma}\hat{c}_{j,\sigma'}|\Psi(t)\rangle$. During the measurement process, the correlation matrix changes according to the following protocol:

1. Extract a random number $p_l \in (0,1]$ for each site $l$ in the top chain. If $p_l \leq p$ then the measurement is performed, otherwise the site is left intact.

2. If the measurement has to be performed, extract a second random number $q_l \in (0,1]$.

3. If $q_l \leq \mathcal{D}_{ll,22} = \langle c^\dagger_{l,2}c_{l,2}\rangle$, then the operator $\hat{n}_{l,2}$ is applied to the state $|\Psi\rangle \to \hat{n}_{l,2}|\Psi\rangle$ which results into the following change of the correlation matrix

$$\mathcal{D}_{ij,\sigma\sigma'} \to \mathcal{D}_{ij,\sigma\sigma'} + \delta_{il}\delta_{jl}\delta_{\sigma2}\delta_{\sigma'2} - \frac{\mathcal{D}_{il,\sigma2}\mathcal{D}_{lj,2\sigma'}}{\mathcal{D}_{ll,22}} . \tag{4}$$

4. If $q_l \geq \mathcal{D}_{ll,22}$, then the operator $1 - \hat{n}_{l,2}$ is applied to the state which results into

$$\mathcal{D}_{ij,\sigma\sigma'} \to \mathcal{D}_{ij,\sigma\sigma'} - \delta_{il}\delta_{jl}\delta_{\sigma2}\delta_{\sigma'2} + \frac{(\delta_{il,\sigma2} - \mathcal{D}_{il,\sigma2})(\delta_{lj,2\sigma'} - \mathcal{D}_{lj,2\sigma'})}{(1 - \mathcal{D}_{ll,22})} . \tag{5}$$

After the measurement process is complete, the cycle of unitary evolution and measurements is repeated for a number of times $t_{st}$.

## 3 Measurement induced transition

To investigate the properties of the entanglement transition, we unravel the dynamics of the system by using the quantum trajectory approach [106–108]. Along each quantum trajectory $\alpha$ the system evolves with a circuit operator $\hat{C}_\alpha$ given by the sequence of unitaries and measurement operations. Different trajectories $\alpha$ and $\alpha'$ differ from each other by the location and time of the measurements. We start from an initial state of the system $|\Psi(0)\rangle$ with a random distribution of particles and let the system evolve along some trajectory $\alpha$. After reaching the steady state $|\Psi^\alpha(t_{st})\rangle = \hat{C}_\alpha|\Psi(0)\rangle$, we calculate the expectation value of some operator $\mathcal{O}$ on the $\alpha^{\text{th}}$ trajectory as $\langle\mathcal{O}^{(\alpha)}\rangle_{t_{st}} = \langle\Psi(0)|\hat{C}^\dagger_\alpha\hat{\mathcal{O}}\hat{C}_\alpha|\Psi(0)\rangle$. For each trajectory, the corresponding observable is averaged over next $m$ timesteps as

$$\langle\langle\mathcal{O}^{(\alpha)}\rangle\rangle_{t_{st}} = \frac{1}{m}\sum_{s=1}^{m}\langle\mathcal{O}^{(\alpha)}\rangle_{t_{st}+s\tau_{\text{u}}} . \tag{6}$$

This process is repeated for $N_{traj}$ trajectories, yielding the steady state trajectory averaged value of operator $\mathcal{O}$

$$\overline{\mathcal{O}} = \frac{1}{N_{traj}}\sum_{\alpha=1}^{N_{traj}}\langle\langle\mathcal{O}^{(\alpha)}\rangle\rangle_{t_{st}} . \tag{7}$$

Throughout this paper, we drop the notation for additional averaging over $t_{avg}$ times and fix $m = 5$ for all observables.

### 3.1 Regime of persistent measurements

We first consider the limiting case of persistent measurements, i.e. $p = 1$, which corresponds to always measuring every site of the outer chain. In such regime, the state of the inner chain is always pure, because at every round of measurements the state is separable as the product of

the state on the outer chain and the state on the inner chain. Thus, upon tracing out the outer chain we obtain a pure state for the inner chain, and we can use the entanglement entropy as a true measure of entanglement within the inner chain, since it only carries quantum correlation and does not include any classical contribution. This is not true anymore in the $p < 1$ case, where the state is not separable, and after integrating out the outer chain degrees of freedom we obtain a mixed state for the inner chain.

Thus in this section, we employ the entanglement entropy and the mutual information as entanglement measures.

When calculating the bipartite entanglement entropy, we choose $A = A_1$ and $A_2 = \varnothing$ (see Fig.(1b) ), and divide the system into $A$ and its corresponding complement segment $\bar{A} = B$. The Von Neumann entanglement entropy of subsystem $A$ is defined as

$$\mathcal{S}_A = -\mathrm{Tr}(\rho_A \log \rho_A), \tag{8}$$

where $\rho_A = \mathrm{Tr}_B \rho_{A \cup B}$, $\mathrm{Tr}_B$ being the trace over the degrees of freedom of the complement subsystem $B$.

The Gaussianity of the state of the system allows us to extract the entanglement properties along some trajectory $\alpha$ at the steady state directly from $\mathcal{D}^{(\alpha)}_{ij,\sigma\sigma'} = \mathcal{D}^{(\alpha)}_{ij,\sigma\sigma'}(t_{st})$:

$$\mathcal{S}^{(\alpha)}_A(t_{st}) = -\sum_{\lambda^{(\alpha)}_A} \left[ \lambda^{(\alpha)}_A \log \lambda^{(\alpha)}_A + \left(1 - \lambda^{(\alpha)}_A\right) \log\left(1 - \lambda^{(\alpha)}_A\right) \right], \tag{9}$$

where $\lambda^{(\alpha)}_A$ are the eigenvalues of the reduced correlation matrix $\mathrm{Tr}_B \mathcal{D}^{(\alpha)}_{ij,\sigma\sigma'}(t_{st})$. In Appendix A we study in detail the convergence of $\mathcal{S}^{(\alpha)}_A$ to the steady state as function of $t_{st}$ for various system sizes.

The trajectory averaged steady state entanglement entropy is:

$$\overline{\mathcal{S}}_A = \frac{1}{N_{traj}} \sum_{\alpha=1}^{N_{traj}} \frac{1}{m} \sum_{s=1}^{m} \mathcal{S}^{(\alpha)}_A(t_{st} + s\tau_{\mathrm{u}}). \tag{10}$$

In Appendix B we study the convergence of $\overline{\mathcal{S}}_A$ as function of the number of trajectories for various system sizes. We find that convergence is typically achieved for $t_{st} = 100\tau_{\mathrm{u}}$ and $N_{traj} = 150$, so that we employ these values throughout the rest of the paper, unless stated otherwise. For simplicity, we set $\tau_{\mathrm{u}} = 1$ for the rest of the paper.

The mutual information between two subsystems $A_1$ and $A_2$ quantifies the correlations between them. It is defined as $\mathcal{I}_{A_1,A_2} = \mathcal{S}_{A_1} + \mathcal{S}_{A_2} - \mathcal{S}_A$, where $A = A_1 \cup A_2$. The quantity of interest is the trajectory averaged steady state mutual information:

$$\overline{\mathcal{I}}_{A_1,A_2} = \overline{\mathcal{S}}_{A_1} + \overline{\mathcal{S}}_{A_2} - \overline{\mathcal{S}}_A, \tag{11}$$

following the definition Eq.(10). Various known scaling forms of the entanglement entropy and of the mutual information are summarized in Table 1.

### 3.1.1 Entanglement entropy

In this subsection we report the numerical results for the entanglement entropy.

Some of the features of $\overline{\mathcal{S}}_A$ can already be understood from the properties of Eq.(3). As already explained in Section 2, at the resonance $t_1 = t_2$, a shift of the transverse tunneling amplitude $t_{12} \to t_{12} + n\pi/\tau_{\mathrm{u}}$ ($n \in Z$) leaves the unitary evolution operator invariant (up to a sign)

$$\hat{U}_k\left(\delta = 0, t_{12} + \frac{n\pi}{\tau_{\mathrm{u}}}\right) = (-1)^n \hat{U}_k(\delta = 0, t_{12}). \tag{12}$$

Table 1: Known scaling behavior of the bipartite entanglement entropy $\mathcal{S}_{L/2}$, and of the mutual information $\mathcal{I}_{L/8}$ and $\mathcal{I}_{L/4}$ as function of $L$ for the area-law phase, for the critical (CFT) phase and for the volume-law phase [36, 57, 89].

|  | $\mathcal{S}_{L/2}(L) \sim$ | $\mathcal{I}_{L/4}(L) \sim$ | $\mathcal{I}_{L/8}(L) \sim$ |
|---|---|---|---|
| Area-law | const$> 0$ | 0 | 0 |
| CFT | $\log(L)$ | const$> 0$ | const$> 0$ |
| Volume-law | $L$ | $L^{1/3}$ | 0 |

We expect the entanglement entropy of the system chain to have the same periodic behavior. The $\sigma^x$ term in $\hat{U}_k$, Eq. (3), is responsible for mixing and entangling the degrees of freedom of the two chains; it vanishes for $t_{12} = n\pi$ and is maximum for $t_{12} = (n + 1/2)\pi$. When the two chains are decoupled, the entanglement in the inner chain grows thanks to the scrambling action of $t_1$ and is insensitive to the measurements in the outer chain. Hence, we expect the non-area-law phase to still persist in the vicinity of $t_{12} = n\pi$ and $t_1 = t_2$. On the contrary, for $t_{12} = \pi/2 + n\pi$ the coupling between the chains is maximum, increasing the sensitivity of the entanglement within the inner chain to measurements in the outer chain. In this regime, the area-law emerges.

These considerations are confirmed by the numerical simulations. We calculate the entanglement entropy $\overline{\mathcal{S}}_{l_A}$ as function of the partition size $l_A$, see Fig.(2a). In all figures, the error bar corresponds to a 95% confidence interval obtained from the distribution of $\mathcal{S}_{l_A}$ over trajectories (see Appendix B); if not visible, it is smaller compared to line width or symbols.

For $t_{12} = \pi/2$ we expect an area-law phase: the entanglement saturates quickly with $l_A$ and shows a nearly flat behavior. On the other hand for $t_{12} = \pi$ we do not expect an area-law and in fact $\overline{\mathcal{S}}_{l_A}$ exhibits a dome shape reminiscent of volume law phases.

In Fig.(2b) we plot a colormap of $\delta\overline{\mathcal{S}} = 1 - \overline{\mathcal{S}}_{L/4}/\overline{\mathcal{S}}_{L/2}$ as function of $t_{12}$ and $t_2$. This quantity vanishes when the entanglement obeys an area-law (because $\overline{\mathcal{S}}_{l_A}$ behaves constantly in $l_A$), while it is non zero for non-area-law phases, and is thus a good indicator to distinguish between the two phases. The periodical structure of area and non-area-law phases, due to the periodicity of Eq.(12), emerges in a very clear way in the colormap.

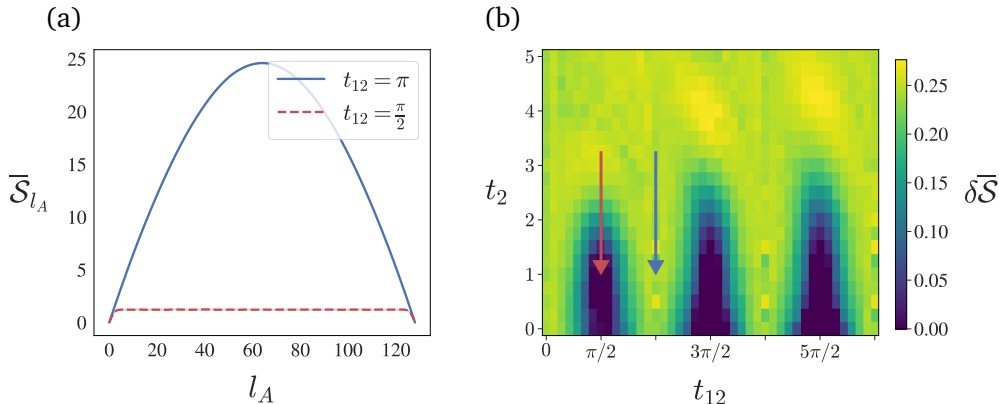

Figure 2: (a) Scaling of $\overline{\mathcal{S}}_{l_A}$ with respect to the partition size $l_A$, for $t_2 = 1$ and $L = 128$. Here the error bar obtained from the 95% confidence interval is smaller than the line width. (b) Colour plot of $\delta\overline{\mathcal{S}} = 1 - \overline{\mathcal{S}}_{L/4}/\overline{\mathcal{S}}_{L/2}$ as a function of $t_{12}$ and $t_2$ with $L = 16$. The red and blue arrows correspond to the values of $t_2$ and $t_{12}$ shown in panel (a), with the corresponding color labeling.

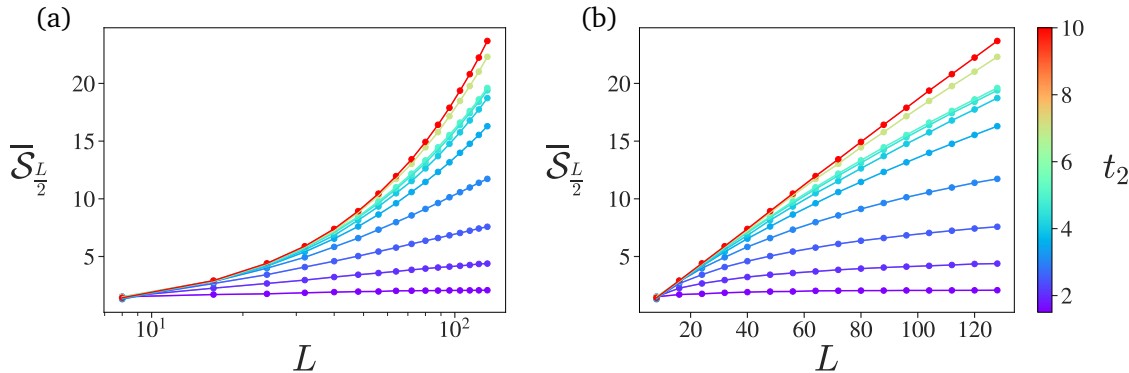

Figure 3: The trajectory averaged entanglement entropy for $l_A = L/2$, for various system sizes and tunneling amplitude $t_2$, with $t_{12} = \pi/2$ fixed, with logarithmic (a) and linear (b) scales.

We now characterize more in detail the non-area law phase. In Table 1 we report the scaling behaviors of the bipartite entanglement entropy $\overline{\mathcal{S}}_{L/2}$, and of the mutual information as function of the system size $L$, for different types of phases. These scaling behaviors can be used to probe the features of the non-area law phase and distinguish between volume-law scaling and critical (CFT) behavior (which has a logarithmic scaling).

We vary $t_2$ along the fixed line $t_{12} = \pi/2$, and study the scaling with $L$ of $\overline{\mathcal{S}}_{L/2}$, which we report in Fig. (3). Based on Fig. (3), we see that the entropy exhibits different behaviors with changing $t_2$. In particular, $\overline{\mathcal{S}}_{L/2}$ displays a clear logarithmic scaling for $t_2 \sim 1.5 \div 3$ [see Fig. 3a)], while its scaling seems more linear (i.e. volume law) in $L$ for large $t_2$, see Fig. 3b)].

In order to quantify the different behaviors, we fit our results with the ansatz

$$\overline{\mathcal{S}}_{L/2} = \gamma L + \frac{c}{3} \log(L) + \beta \,, \tag{13}$$

within a range $L_{\min} \leq L \leq L_{\max}$, for different ranges $[L_{\min}, L_{\max}]$. We calculate $\gamma$ and $c$ and compare $\gamma L_{\max}$ with $c/3 \ln L_{\max}$. We observe a crossover between the logarithmic contribution (dominant at small $t_2$) and the linear contribution, which dominates at larger $t_2$ but is very small below a threshold value of $t_2$. The detailed results and plots are reported in Appendix C. However, as we increase for $L_{\max}$, both the threshold for $\gamma$ and the crossover value of $t_2$ increase, suggesting that the logarithmic scaling is the dominant contribution in the thermodynamic limit and that the presence of linear corrections is a finite size effect.

In order to assess whether this is an artifact of boundary effects from which the entanglement entropy suffers, we also consider the mutual information, where finite size effects are less prominent.

### 3.1.2 Mutual information

In this subsection, we study the mutual information to further investigate the non-area regime.

We look at the behavior of $\overline{\mathcal{I}}_{L/4}$, i.e. the mutual information between diametrically opposing subsystems $A_1$ and $A_2$ with $l_{A_1} = l_{A_2} = L/4$, and plot its behavior in Fig. (4a). Comparing Table 1 and Fig. (4a), the behavior of $\overline{\mathcal{I}}_{L/4}$ is clearly consistent with an area law for $t_2 = 1.5$ and with a CFT phase for $t_2 \lesssim 3.0$. At larger $t_2$, $\overline{\mathcal{I}}_{L/4}$ does not saturate to a constant, but keeps increasing. This is compatible with a CFT phase for which the saturation size of $\overline{\mathcal{I}}_{L/4}$ is larger than the sizes we can access, but does not completely exclude a volume law phase for which $\overline{\mathcal{I}}_{L/4} \sim L^{1/3}$.

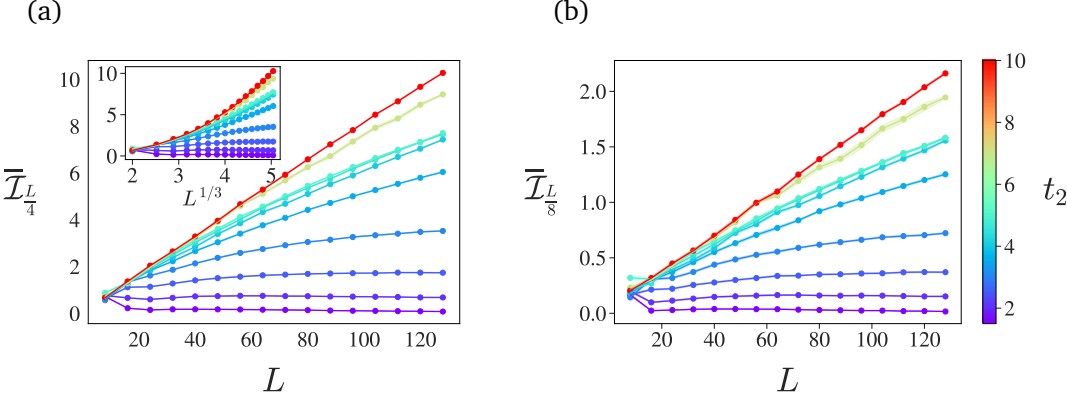

Figure 4: The trajectory averaged mutual information for various system sizes and $t_2$, for $l_A = l_B = l$, with $l = L/4$ for (a) and $l = L/8$ for (b), with fixed $t_{12} = \pi/2$. For the sake of comparison with expected volume law scaling, the inset on panel (a) shows the scaling of $\overline{\mathcal{I}}_{L/4}$ on $L^{1/3}$ scale.

Another useful witness of a phase transition is the behavior of $\overline{\mathcal{I}}_{L/8}$, i.e. the mutual information between diametrically opposing subsystems $A_1$ and $A_2$ with $l_{A_1} = l_{A_2} = L/8$. $\overline{\mathcal{I}}_{L/8}$ vanishes in the area and volume law phases, and is enhanced at the critical point, due to long-range correlations developing between the subsystems [36, 39]. As shown in Fig.(4b), $\overline{\mathcal{I}}_{L/8}$ is very close to zero for $t_2 = 1.5$, indicating an area law. For $t_2 \lesssim 3.0$, $\overline{\mathcal{I}}_{L/8}$ saturates to a constant values, thus agreeing with the presence of a CFT phase. Similarly to $\overline{\mathcal{I}}_{L/4}$, the value of $\overline{\mathcal{I}}_{L/8}$ keeps increasing as $t_2$ gets larger. This is not compatible with a volume law phase, suggesting that also at large $t_2$ the system is in a CFT phase, but due to stronger finite size effects the entanglement entropy displays a linear scaling contribution and the mutual information saturates to values of $L$ larger than the system sizes we can access.

In order to better discriminate the presence of an underlying CFT description of the phase, we study the dependence of the mutual information on the cross ratio $\eta$. Suppose that the system of fixed size $L$ is bipartitioned into two subsystems of length $l_A$ and $l_B$, with the boundaries of the $l_A$ segment located at sites $x_1$ and $x_2$ and the boundaries of segment $l_B$ located at $x_3$ and $x_4$. The cross ratio for such a bipartition is defined as $\eta = \frac{x_{12}x_{34}}{x_{13}x_{24}}$, with $x_{ij} = L/\pi \sin(\pi|x_i - x_j|/L)$. In a CFT regime, the mutual information collapses onto a single line and for small cross ratios shows a power-law growth, i.e. $\mathcal{I}(\eta) \sim \eta^{\Delta}$.

The inset in Fig.(5a) shows the behavior of the trajectory averaged steady state mutual information with respect to $\eta$ for $L = 64$. The data points for $\eta \ll 1$ collapse onto a single line for $t_2 = 1.5$ and for $t_2 = 3.0$, but in the first case they show a larger spread.

In order to reduce the fluctuations due to the spread of the data points and perform a fit for $\Delta$, we restrict our analysis to the special case of diametrically opposite segments $|A| = |B| \sim \sqrt{\eta}L$, for which $I_{A,B} \sim \eta^{\Delta}$. The fitted data for $L = 128$ in Fig.(5a) would suggest that the $t_2 = 3$ and $t_2 = 5$ regimes correspond to a CFT. We observe that for these regimes, the scaling exponent $\Delta$ is very close to 1. For $t_2 = 1.5$ the data points have larger deviation from the $\eta^{\Delta}$ curve at larger $\eta$, meaning that the fit for $\Delta$ is not very reliable. In order to obtain a more precise results in the latter case, computationally costly simulations with a larger number of trajectories are needed. By calculating $\Delta$ for various $t_2$ and system sizes Fig.(5b), we see that $\Delta \to 1$ for the regimes far away from the area-law phases ($t_2 = 3$ and 5).

Thus our analysis points to the presence of a CFT phase for all values of $t_2 \geq 1.5$. The linear scaling of the entanglement entropy at large $t_2$ seems to be a finite size effect, since the presence of a volume law phase is in contradiction with the large long-range correlations

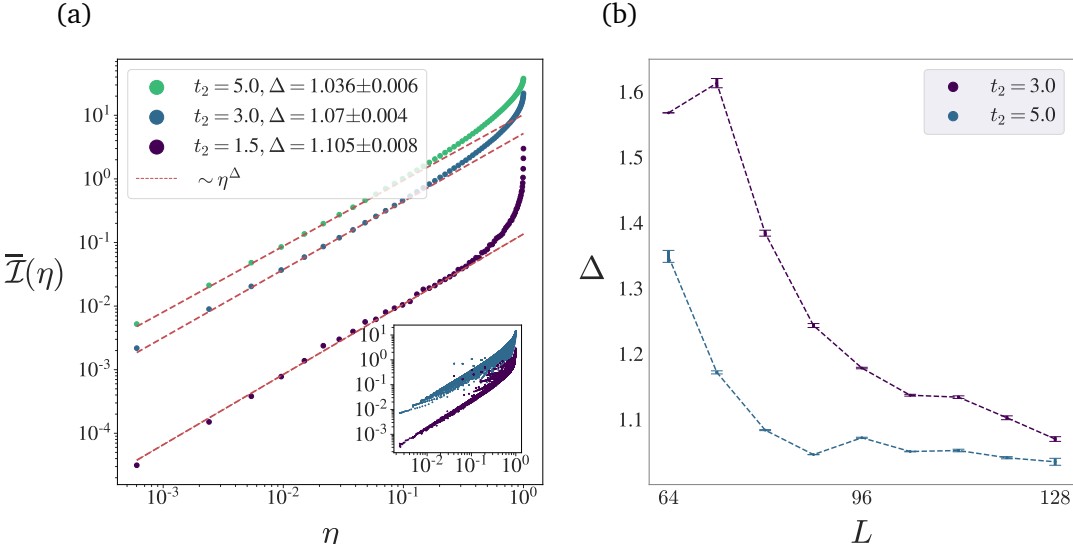

Figure 5: (a) Trajectory averaged mutual information with $L = 128$, with fixed $t_{12} = \pi/2$ and $t_2 = 1.5, 3$ and $5$, for $l_A = l_B$ and $r_{AB} = L/2$. The red dashed lines correspond to $\sim \eta^\Delta$ fit along the data-points. The fitting curve is $I(\eta) = a(e^{b\eta^c} - 1)^d$, which for $\eta \ll 1$ reduces to $I(\eta) \approx ab^d \eta^{cd}$ and thus $\Delta = cd$. The inset shows the mutual information for $L = 64$, for every tripartitions, with $t_{12} = \pi/2$ and $t_2 = 1.5$ and $3$. The collapse of the data points onto a single curve is evident for smaller values of $\eta$ and $t_2 = 1.5$, while there is a significant spread at larger $\eta$ or for $t_2 = 3$. (b) The scaling behavior of $\Delta$ exponent, for $t_{12} = \pi/2$ and $t_2 = 3$ and $5$.

indicated by the large values of $\overline{\mathcal{I}}_{L/8}$. This behavior is reminiscent of what found in [57], where the entanglement entropy displays linear finite size corrections below a saturation size $L$ that depends on the measurement rate.

However, without access to larger system sizes we cannot definitively exclude a volume law phase. We point out that for a single free fermion chain, the underlying unitary dynamics is expected to contribute logarithmically [46] – indeed the ground state of the unmonitored chain is a CFT phase – so that one would expect to observe an area-to-log transition. This is no longer guaranteed for a ladder, where the entanglement entropy within one leg may scale linearly with size.

We also remark that the presence of bigger long-range correlations at larger $t_2$ makes intuitively sense. In fact, when $t_2$ is large, information spreading in the outer chain is fast, meaning that a local measurement can still affect the neighboring sites, and the range of this effect increases with $t_2$. In other words, the time correlations between measurements due to the internal dynamics of the bath chain translate into space correlations at the level of the system chain. In a sense, this can be interpreted as non-Markovian effects inducing long-range correlations within the system chain.

## 3.2 Regime of sporadic measurements

For measurement probabilities that are smaller than one, the Von Neumann entropy and the mutual information are not valid measures of quantum entanglement, since they also include classical correlations, given that the reduced state of the inner chain is now mixed.

To properly quantify the entanglement properties of the inner chain, we use the logarithmic fermionic negativity [88, 92, 93], not to be confused with the logarithmic negativity, which is used for systems of commuting particles. Negativity is an entanglement monotone for mixed

states, whose bosonic version has already found use in the context of measurement induced transitions [90, 91].

Suppose the inner chain, described by the corresponding reduced density matrix $\rho_{\text{sys}}$, is further bipartitioned into $A$ and $B$ (as in Fig.(1a) with $A = A_1$ and $A_2 = \varnothing$). The logarithmic fermionic negativity $\mathcal{E}_A$ of subsystem $A$ is defined as

$$\mathcal{E}_A = \log \text{Tr} |\rho_{\text{sys}}^{\tilde{T}_A}|, \tag{14}$$

with

$$\rho_{\text{sys}}^{\tilde{T}_A} = \rho_{\text{sys}}^{T_A} (-1)^{F_A}, \tag{15}$$

where $\rho_{\text{sys}}^{\tilde{T}_A}$ and $\rho_{\text{sys}}^{T_A}$ are the twisted and untwisted partial transpose of the reduced density matrix, respectively. Both operations are performed only on the $A$ sub-system, leaving $B$ intact. $F_A$ is the number of fermions in the sub-system $A$.

Thanks to the Gaussianity of the state, the fermionic logarithmic negativity can be extracted from the correlation matrix $\mathcal{D}$ [109, 110]. Since in our model $\langle c_{j,\sigma}^\dagger c_{j',\sigma'}^\dagger \rangle = 0$ for all times, the computation of the negativity is simpler than in Ref. [92] and the passage to the Majorana fermions can be avoided. Given a reduced correlation matrix $\mathcal{D}_{\text{sys}}$ of an entire inner chain, we define $(\Gamma_{\text{sys}})_{ij,\sigma\sigma'} = 2(\mathcal{D}_{\text{sys}})_{ij,\sigma\sigma'} - \delta_{ij}\delta_{\sigma\sigma'}$. For a bipartition of the inner chain to $A$ and $B$, the $(\Gamma_{\text{sys}})_{ij,\sigma\sigma'}$ matrix is expressed as

$$\Gamma_{\text{sys}} = \begin{pmatrix} \Gamma_{AA} & \Gamma_{AB} \\ \Gamma_{BA} & \Gamma_{BB} \end{pmatrix}, \tag{16}$$

where each block corresponds to the correlations between the segments indicated in the subscript. From the block structure of $\Gamma_{\text{sys}}$, we introduce the transformed matrices

$$\Gamma_\pm = \begin{pmatrix} \Gamma_{AA} & \pm i\Gamma_{AB} \\ \pm i\Gamma_{BA} & -\Gamma_{BB} \end{pmatrix}, \tag{17}$$

and

$$\Gamma_* = \frac{1}{2}\left[1 - (1 + \Gamma_+\Gamma_-)^{-1}(\Gamma_+ + \Gamma_-)\right]. \tag{18}$$

Using these matrices, the calculation of the Fermionic negativity is straightforward

$$\mathcal{E}_{A_1} = \sum_j \left[\ln\left(\sqrt{\mu_j} + \sqrt{1-\mu_j}\right) + \frac{1}{2}\ln\left(1 - 2\lambda_j + 2\lambda_j^2\right)\right], \tag{19}$$

where $\mu_j$ and $\lambda_j$ are the eigenvalues of $\Gamma_*$ and $\mathcal{D}_{\text{sys}}$, respectively.

We calculate the logarithmic fermionic negativity for the steady state and average it over different trajectories, yielding $\overline{\mathcal{E}}_A$.

We have analyzed the scaling properties of the fermionic negativity, for fixed system size $L = 64$ and different values of $p$ and $t_2$, shown in Fig.(6). Figure (6a) corresponds to the fine-tuned resonance regime $t_2 = 1$ and $t_{12} = \pi/2$. The peak of the negativity at $l_A = L/2$ is reduced as the measurement probability $p$ increases and the curve is progressively flattened out. This behaviour is expected, since for $t_1 = t_2$ and $t_{12} = \pi/2$ the two chains exhibit maximum coupling and thus, the entanglement content within the inner chain is reduced when the outer chain is frequently measured. The flattening of $\overline{\mathcal{E}}_{l_A}$ for larger $p$ indicates the onset of the area-law regime.

Figure (6b) corresponds to the case of parameters far away from the resonant regime. Contrary to the previous case, now the peak of the negativity becomes larger as $p$ increases. This surprisingly counterintuitive behavior can be explained in terms of entanglement monogamy

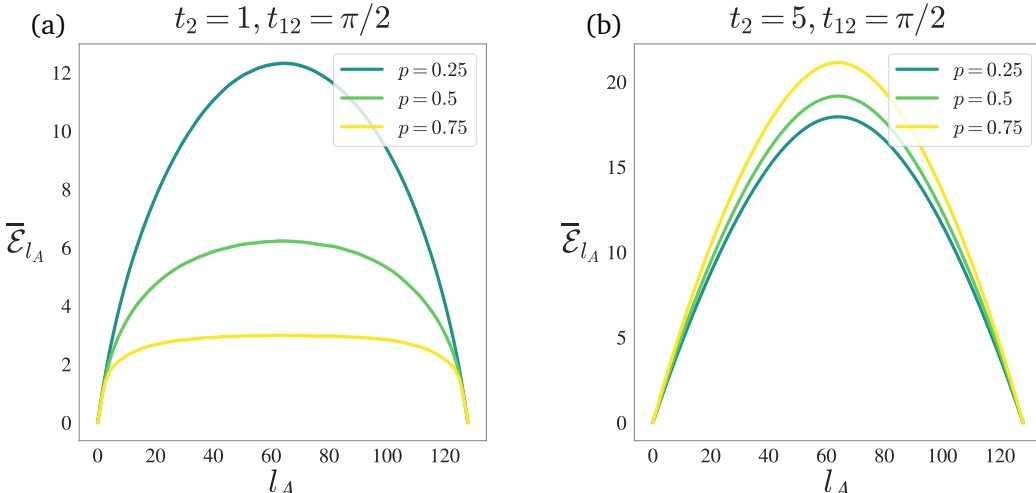

Figure 6: Figures show the scaling of $\overline{\mathcal{E}}_{l_A}$ with respect to $l_A$ with $L = 64$, for various values of $p$.

[111–113]. The relatively large value of $t_2$ quickly spreads the entanglement throughout the outer chain, before the next measurement occurs, while the coupling between the degrees of freedom of the inner and outer chains, is maximal. For low measurement probability, more entanglement is spread within the outer chain and between the outer chain and the inner chain, leaving less entanglement available to be distributed within two partitions of the inner chain, which thus shows a lower value of the fermionic negativity. This occurs because the total amount of entanglement that can be shared within a tripartite system (in our case the outer chain and two partitions of the inner chain) is limited, so that the more information two subsystems share with the third subsystems, the less information they will share after tracing it out. Conversely for large $p$, the entanglement between the two chains is smaller due to frequent measurements. This means less information is discarded when performing the partial trace over the outer chain, and more entanglement content is shared within the inner chain.

Similarly to what was done for the entanglement entropy in Fig.(2), we extract the phase diagram of the system for $p < 1$. We characterize the phases using $\delta\overline{\mathcal{E}} = 1 - \overline{\mathcal{E}}_{L/4}/\overline{\mathcal{E}}_{L/2}$ as function of $t_{12}$ and $t_2$, and find that again a periodic structure of the phase diagram emerges Fig.(7). For smaller $p$, the faint remnants of the area-law lobes are still present. As $p$ is increased, these lobes become more prominent and visible.

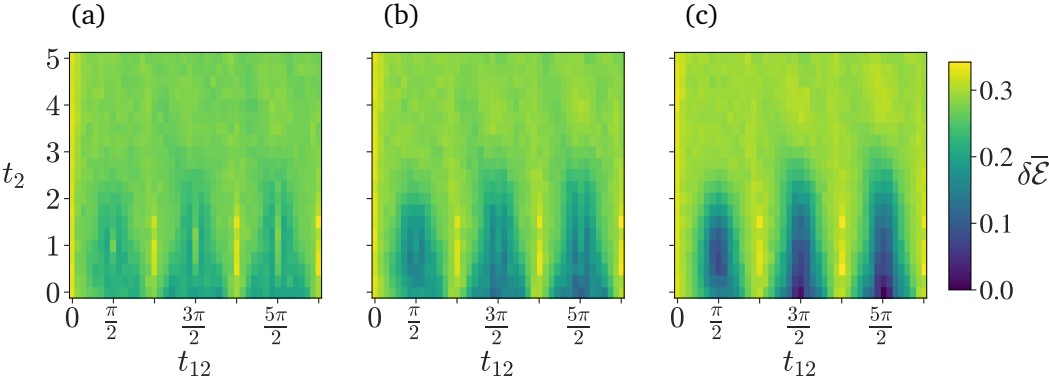

Figure 7: Color-maps of the negativity difference $\delta\overline{\mathcal{E}} = 1 - \overline{\mathcal{E}}_{L/4}/\overline{\mathcal{E}}_{L/2}$ for $L = 16$, with $p = 0.25$ in panel (a), $p = 0.5$ in panel (b) and $p = 0.75$ in panel (c).

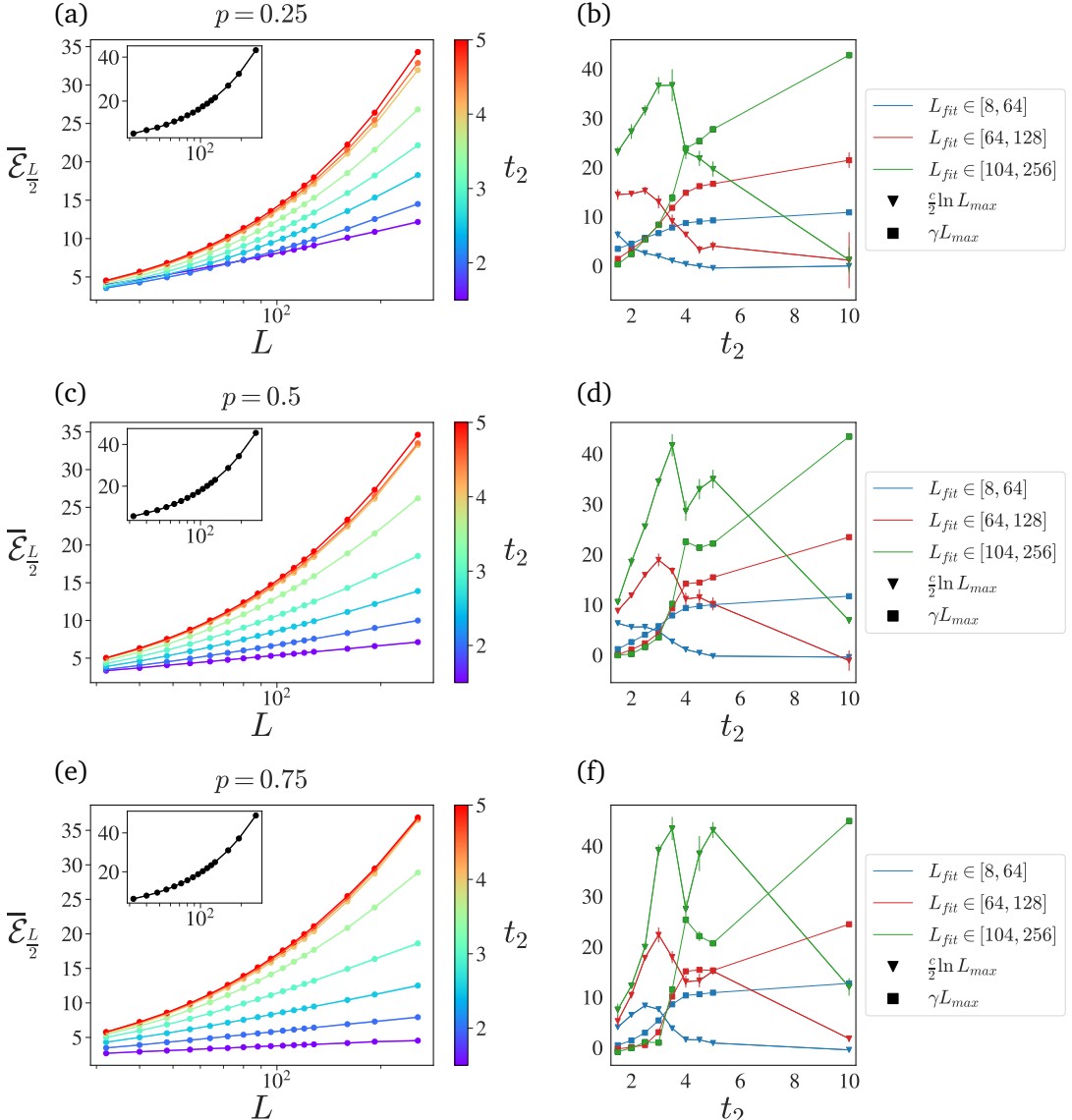

Figure 8: Panels (a,c,e) correspond to the trajectory averaged fermionic negativity for $l_A = L/2$, for various system sizes, tunneling amplitude $t_2$ and measurement probability $p$. The value of the inter-chain tunneling amplitude is fixed to $t_{12} = \pi/2$. The insets correspond to $t_2 = 10$ regime. Panels (b,d,f) are the plots of $\gamma L_{max}$ and $c/2\ln(L_{max})$ versus $t_2$ tunneling amplitudes. The points is extracted from fitting the data along $\overline{\mathcal{E}}_{L/2} = c/2\ln(L) + \gamma L + \beta$ curve.

We then study the scaling of $\overline{\mathcal{E}}_{L/2}$ for various values of $p$ and $t_2$, which shows different scaling properties. For $p = 0.75$ we clearly see an area law behavior at small $t_2$ and a logarithmic behavior at larger $t_2$; the area law at $t_2 = 1.5$ disappears for $p = 0.25$, in agreement with Fig. 7. We also observe that for large $t_2 \sim 5 \div 10$, the negativity exhibits a linear behavior at small sizes and a seemingly logarithmic behavior at large sizes $L \gtrsim 128$. To confirm this claim, we fit the data along $\overline{\mathcal{E}}_{L/2} = c/2\ln(L) + \gamma L + \beta$ [92] curve and extract the corresponding coefficients. Figures (8b,d,f) show the behavior of $\gamma L_{max}$ and $c/2\ln(L_{max})$ for different fitting ranges (with $L_{max}$ the maximum size of each range). The plot show that the logarithmic contribution clearly dominates for small $t_2$, while it is comparable with the linear term for large $t_2$; however, increasing the sizes in the fitting range, the logarithmic contribution increases

faster than the linear one at all values of $t_2$. This suggests that the linear contribution is a finite size effect, which is stronger at small $L$ and large $t_2$ but becomes negligible as larger and larger system sizes are considered. This behavior is qualitatively similar to what we found for the $p = 1$ case, confirming the rich variety of phases present in this model.

The trend of the logarithmic and linear contributions with the range of the fit and an analogy with the behavior of the entanglement entropy, suggest that the linear contribution at large $t_2$ is due to finite size effects. However we cannot make a definitive claim and exclude completely a volume law phase, since we only have the negativity at our disposal to quantify the entanglement, and cannot cross-check it with a second observable such as the mutual information for the $p = 1$ case.

## 4  Measures of non-Markovianity

In this Section we assess how non-Markovian the dynamics of the inner chain is, and we show that the model we study is indeed a good platform to simulate non-Markovian systems.

In order to quantify the degree of non-Markovianity, we use the measure $\mathcal{N}(\Phi)$ defined in Ref. [95], where $\Phi$ is a dynamical map acting in the space of density matrices such that $\Phi : \rho(0) \to \rho(t) = \Phi(t)\rho(0)$. Here $\Phi$ may represent for example the map generated by a master equation $\dot{\rho} = \mathcal{L}\rho$, such that $\Phi(t) = e^{\mathcal{L}t}$.

We first define the trace distance between two density matrices as

$$d_\rho(\rho_1, \rho_2) \equiv \frac{1}{2}\text{Tr}|\rho_1 - \rho_2|, \qquad |\rho| = \sqrt{\rho^\dagger \rho}. \tag{20}$$

It can be shown that the trace distance between two density matrices $\rho_1$ and $\rho_2$ always decreases in time for Markovian maps, i.e. $d_\rho(\Phi\rho_1, \Phi\rho_2) \leq d_\rho(\rho_1, \rho_2)$. It is then natural to distinguish Markovian from non-Markovian dynamics based on whether $d_\rho$ always decreases or may also increase, and to quantify the degree of non-Markovianity of a map $\Phi$ with how much the distance between two density matrices increases over time. Following Ref. [95], we define the time derivative $\sigma_\Phi$ at time $t$ of the trace distance for a given map, and for two initial density matrices $\rho_1(0)$ and $\rho_2(0)$, as

$$\sigma_\Phi(t, \rho_{1,2}(0)) = \frac{d}{dt}d_\rho(\Phi(t)\rho_1(0), \Phi(t)\rho_2(0)). \tag{21}$$

For a Markovian map $\sigma_\Phi$ is always negative, while it may become positive for finite time intervals for a non-Markovian map.

The non-Markovianity measure $\mathcal{N}(\Phi)$ is then defined as the maximum over all possible initial conditions of the integral of $\sigma_\Phi$ over the times where it is positive

$$\mathcal{N}(\Phi) = \max_{\rho_{1,2}(0)} \int_{\sigma_\Phi > 0} dt\, \sigma_\Phi(t, \rho_{1,2}(0)). \tag{22}$$

The calculation of $\mathcal{N}(\Phi)$ is demanding, since it involves calculating the distance between density matrices, and the Gaussian state formalism employed in the previous sections cannot be applied. Moreover, calculating the maximum over the pairs of initial density matrices, means sampling a space whose dimension scales exponentially with the size of the system.

We use exact diagonalization (ED) techniques to numerically simulate the dynamics of the model defined in Section 2. We calculate the evolution of the total density matrix of the two chains according to the Lindblad master equation that results from averaging over trajectories Eqs. (3)-(5). Since we already calculate the average dynamics, we do not need to perform the evolution over different trajectories and then average. We choose $L = 4$ and for each time step

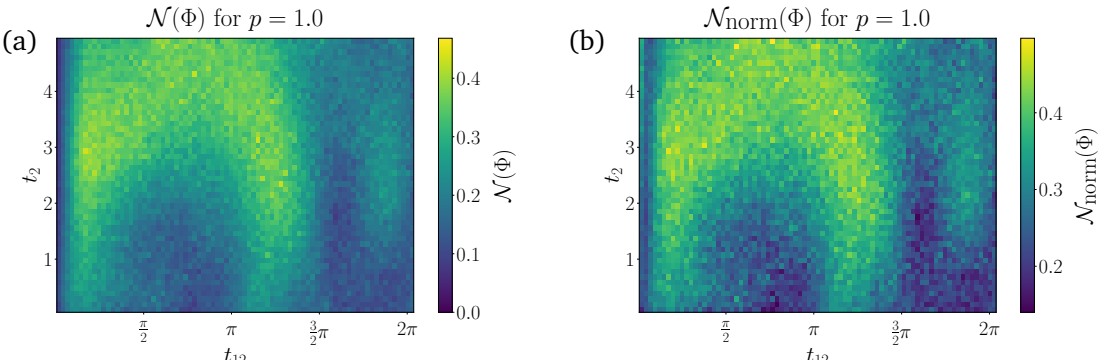

Figure 9: (a) Colormap of the non-Markovianity measure $\mathcal{N}(\Phi)$ for the process defined by the map $\Phi(t)$, as function of $t_2$ and $t_{12}$ for probability measurement $p = 1$ and for system size $L = 4$. (b) Colormap of the normalized non-Markovianity measure $\mathcal{N}_{\mathrm{norm}}(\Phi)$ calculated for the same parameters of (a).

we trace out the outer chain in order to obtain the density matrix of the inner chain as function of time. We sample over a number of pairs of random initial density matrices $N_{\mathrm{pairs}} \sim 100$. This sampling is the largest source of fluctuations in our simulations: especially for small values of non-Markovianity, the number of initial pairs that we need to sample to get a non zero value is rather large – as it scales exponentially with the size of the system.

Our results are reported in Fig. 9 and 10. In Fig. 9 we plot the color maps of $\mathcal{N}(\Phi)$ and $\mathcal{N}_{\mathrm{norm}}(\Phi)$ as function of $t_2$ and $t_{12}$ for $p = 1$. Here we have defined the normalized non-Markovianity measure $\mathcal{N}_{\mathrm{norm}}(\Phi)$ by dividing the integral over the regions of non-Markovianity with the integral over time of $|\sigma|$ and then maximizing over the pairs of initial density matrices, i.e.

$$\mathcal{N}_{\mathrm{norm}}(\Phi) = \max_{\rho_{1,2}(0)} \frac{\int_{\sigma_\Phi > 0} dt\, \sigma_\Phi(t, \rho_{1,2}(0))}{\int dt\, |\sigma_\Phi(t, \rho_{1,2}(0))|} \,. \tag{23}$$

The integral of $|\sigma|$ is typically of order one, so that $\mathcal{N}(\Phi)$ and $\mathcal{N}_{\mathrm{norm}}(\Phi)$ have usually the same order of magnitude. However, the normalized measure is still useful to identify non-Markovianity in regimes where the decay of the density matrix towards its equilibrium value is slow, such as in the small $t_{12}$ regime, see Fig. 9(b).

We find that the dynamics of the inner chain is always non-Markovian. The degree of non-Markovianity is not uniform in the $t_{12}$-$t_2$ plane, and $\mathcal{N}(\Phi)$ exhibits a behavior similar to that of the entanglement entropy and of the negativity. Except for the region of very small $t_{12}$, where

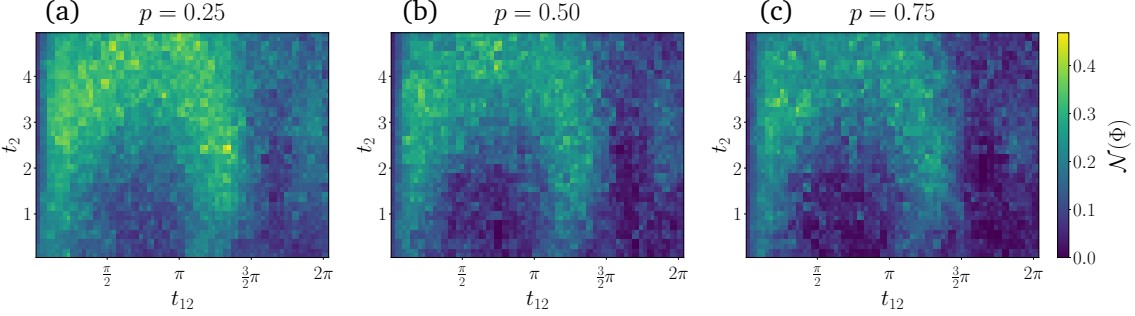

Figure 10: Colormap of the non-Markovianity measure $\mathcal{N}(\Phi)$, as function of $t_2$ and $t_{12}$, calculated for system size $L = 4$ and for measurement probabilities $p = 0.25$ (a), $p = 0.5$ (b), $p = 0.75$ (c). The color scale is the same as Fig. 9(a).

we expect $\mathcal{N}(\Phi)$ to be small due to the weak interchain coupling, the regions of strong non-Markovianity coincide qualitatively with the regions of non-area-law entanglement, see Fig. 2. This makes intuitive sense; for example, at $p = 1$ one would always expect an area law, but for large $t_2$ this does not occur because the internal dynamics of the bath (outer chain) scrambles the effects of the measurements, meaning that there are strong non-Markovian effects.[2] This is indeed reflected in the behavior of $\mathcal{N}(\Phi)$. The phase diagrams of Fig. 2 and Fig. 9 do not coincide exactly, since the periodicity of $\mathcal{N}(\Phi)$ in $t_{12}$ seems to be larger than $\pi$ as found for entanglement. This is likely due to finite size effects, given the very small system size we use to calculate $\mathcal{N}(\Phi)$.

We also calculate $\mathcal{N}(\Phi)$ for $p = 0.25, 0.5, 0.75$, see Fig. 10. We do not find any qualitative difference with the $p = 1$ case. We observe that the degree of non-Markovianity is generally smaller for $p = 0.5$ and $p = 0.75$, while it increases again for $p = 0.25$. This non-monotonous behavior of $\mathcal{N}(\Phi)$ as function of $p$ is akin to what we found in Fig. 6(b) because of entanglement monogamy. However, given the large fluctuations and the small system sizes of these simulations, we do not have enough accuracy to speculate further on the meaning of this result.

Note that we need to use ED techniques to calculate $d$ since we cannot take advantage of the Gaussianity of the state. In fact the average density matrix $\rho(t)$ of the two chains is obtained as the average over trajectories of the pure (Gaussian) state density matrices $\rho(t) = \sum_\alpha |\psi(t)\rangle \langle\psi(t)|^{(\alpha)} / N_{traj}$, and Gaussianity is not an additive property so that $\rho(t)$ cannot be expressed as a Gaussian state. The same problem is encountered if the relative entropy $d_{\log}(\rho_1, \rho_2) \equiv \text{Tr}(\rho_1(\log\rho_1 - \log\rho_2))$ is used as trace distance, since $\log\rho_1$ cannot be written as a Gaussian state. On the other hand, one can take advantage of the Gaussianity of the state if the $\mathbb{L}_2$ quadratic trace distance [114] is used:

$$d_2(\rho_1, \rho_2) = \sqrt{\text{Tr}|\rho_1 - \rho_2|^2}. \tag{24}$$

In fact $\text{Tr}(\rho_1^2) = \text{Tr}\sum_{\alpha,\alpha'} |\psi_1\rangle \langle\psi_1|^{(\alpha)} |\psi_1\rangle \langle\psi_1|^{(\alpha')} / N_{traj}^2$, i.e. it is a double average of the product of two Gaussian states, which is still a Gaussian state whose trace can be calculated in terms of the two-points correlation matrix [85]. The same is true for $\rho_2^2$ and $\rho_1\rho_2$, so that $\text{Tr}|\rho_1 - \rho_2|^2$ can be calculated from the behavior of the correlation matrix averaged twice over trajectories.

In order to evaluate the degree of non-Markovianity we will need to maximize it over different pairs of initial conditions $\rho_1$ and $\rho_2$, i.e.

$$\mathcal{N} = \max_{1,2} \int_{\partial_t d_2 > 0} dt \, \partial_t d_2(t). \tag{25}$$

Though the calculation of $d_2(\rho_1, \rho_2)$ from the two-point correlation function of the system $\mathcal{D}_{sys} = \text{Tr}_{\sigma=2}\mathcal{D}$ is straightforward, it comes with a large computational cost. In fact, while all the operations to be performed have a polynomial cost in the system size, there is a large overhead originating from the double average over trajectories, which is completely absent for ED numerics. Moreover, the calculation of the product of two Gaussian states in each of the terms of the double average, requires inverting the correlation matrices, an operation that becomes expensive when the system size increases. This actually makes calculations of $d_2(\rho_1, \rho_2)$ rather expensive for large system sizes. For this reason, we have restricted ourselves with $L = 8$ system size, with $N_{pairs} \sim 300$ number of pairs of initial conditions and $N_{traj} = 50$ trajectories per initial condition. The total running time for a single trajectory is fixed to $T_{max} = 100$ time-steps.

---

[2]We point out that this is not true for arbitrary large $t_2$; for $t_2 \to \infty$ one expects the dynamics of the bath to be so fast that it results in a Markovian behavior.

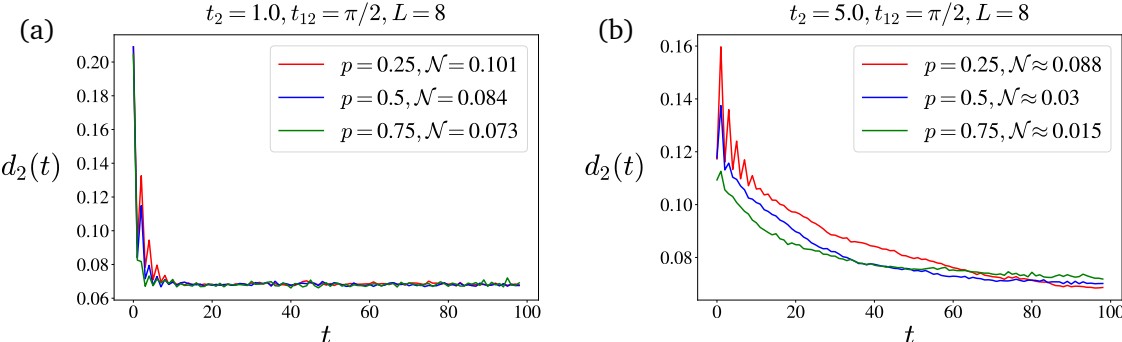

Figure 11: The time evolution of quadratic trace distance $d_2(\rho_1, \rho_2)$, for various values of $t_2$ and measurement probability $p$, with fixed $t_{12} = \pi/2$ and $L = 8$. The curves correspond to the pair initial conditions $\rho_1$ and $\rho_2$ that maximize the non-Markovianity measure $\mathcal{N}$.

We present the results in Fig. 11, where the non-Markovianity of the dynamics (increasing trace distance) can be clearly seen. Similarly to ED simulations, for $t_2 = 5$ we observe that the degree of non-Markovianity is enhanced for smaller values of $p$. For a fine-tuned regime $t_2 = t_1 = 1$, the degree of non-Markovianity increases with increasing $p$. It should be noted, that the trace distance does not saturate to zero, likely due to small Hilbert space size of the model with $L = 8$. Moreover, we see that for $p = 0.5$ and $p = 0.75$, 11 shows that the degree of non-Markovianity is larger for $t_2 = 1$ and smaller for $t_2 = 5$, however the opposite trend is visible on Figs. 10b,c. These differences allows us to only conclude that the dynamics is always non-Markovian and do not allow us to make any further statements regarding the degree of non-Markovianity for various regimes of tunneling amplitudes $t_{2,12}$ and measurement probability $p$.

A third (hybrid) method that could be used consists in simulating the time evolution of the correlation matrix using Gaussian states, calculating the corresponding density matrix for each trajectory and time step, and averaging them over trajectories to obtain the average density matrix given a certain initial condition. However, this method results more costly than using only Gaussian states and performing all necessary calculations on the correlation matrices, since it involves calculating the exponential of a matrix of size $2^{2L} \times 2^{2L}$ which scales like a matrix multiplication and thus $\sim (2^{2L})^3$. A brief comparison of the computational complexity of each method is reported in Table 2.

Table 2: Scaling of the computational complexity to calculate $\mathcal{N}(\Phi)$ for different methods. $a$ is the number of matrix multiplication performed in one time step during the ED simulations (typically $a \sim 3$), $t_{st}$ is the number of time steps, $N_{pairs}$ is the number of different initial conditions and $N_{traj}$ is the number of trajectories. $2^{6L}$ is the typical computational cost of the multiplication of the density matrix of a system of size $2L$.

| Method | ED | Hybrid (Gaussian+ED) | Gaussian |
|---|---|---|---|
| Complexity | $a2^{6L}t_{st}N_{pairs}$ | $((2L)^3 + 2^{6L})t_{st}N_{traj}N_{pairs}$ | $(2L)^3 t_{st}N_{traj}N_{pairs}$ |

# 5 Conclusions

In this paper we have studied a model of ladder of free fermions with periodic boundary conditions. The inner chain of the ladder acts as the system under study, while the outer chain acts as an environment with an internal dynamics. Measurements are performed on the outer chain, thus inducing an effective non-Markovian dynamics on the inner chain once the outer chain is traced out.

We have investigated the entanglement transition in this system by analyzing several witnesses of entanglement within the inner chain. More specifically, we studied the bipartite entanglement entropy and the mutual information between diametrically opposite partitions when the outer chain is always measured – i.e. when $p = 1$ and the the outer chain is in a pure state – and the fermionic negativity for $p < 1$.

We analyzed the phase diagram as function of the hopping between the chains ($t_{12}$) and within the bath chain ($t_2$) at $p = 1$, and were able to distinguish between area-law phase and non-area-law phase by looking at the scale with partition size of the entanglement entropy at fixed $L$. We found that the non-area law survives even for strong measurements when the scrambling rate of the outer chain (which is proportional to $t_2$) is large enough. We also found a periodic structure in $t_{12}$ of the phase diagram, with repeating lobes of area-law phases, which arises due to the geometric structure of the ladder.

For a value of $t_{12}$ that maximizes the interchain coupling, we investigated the nature of the non-area law phase to understand whether it is a volume-law phase or a conformally invariant phase. We studied the scaling with system size $L$ of the bipartite entanglement entropy and of the mutual information. From the data of the entanglement entropy, we found that it scales logarithmically at smaller $t_2$, near the values of $t_2$ that give raise to the area-law phase. For larger values of $t_2$ in between, the entropy does not show one distinct behavior but a mix of logarithmic and volume law scaling.

The analysis of the mutual information clearly confirms the presence of the CFT phase at smaller values of $t_2$, close to the lobes of area-law phases, although larger system sizes would be needed for a definitive assessment. The behavior of the mutual information also indicates that the system possesses a large amount of long-range correlations even when the entanglement entropy seems to scale linearly. This behavior is not compatible with a volume-law phase, for which the mutual information would vanish, leading us to conclude that the system is in a CFT phase and the linear corrections to the scaling of the entanglement entropy are due to finite size effects. The presence of big long-range correlations is explained by the large hopping value of $t_2$, which favors the creation of long-range entanglement in the system chain.

A precision study of the transition could be performed, for example by integrating our analysis with additional diagnostic tools. One useful observable is for example the tripartite mutual information, whose behavior is able to precisely pinpoint the transition. Such studies are however beyond the scope of this manuscript, and we reserve them for future works.

We also considered the case of $p < 1$ and studied the entanglement transition using the fermionic negativity. We found a qualitatively similar behavior to the $p = 1$ case. In the regime of large $t_2$ the scaling of the negativity is dominated by a volume law for all values of $p$ we consider. For smaller $t_2$, the logarithmic contribution to the scaling dominates and the phase is CFT-like, although the width of the window in which this occurs shrinks for smaller values of $p$. A striking result is that for large $t_2$, the entanglement in the inner chain can be increased by performing more measurements on the outer chain, a phenomenon which we postulate is due to the monogamy of entanglement. Although the behavior of negativity is very similar to that of the entanglement entropy, we cannot in principle make the claim that the linear contributions are due to finite size effects, as we lack the $p < 1$ counterpart of the mutual

information to precisely pinpoint the phase of the system.

A remarkably similar phenomenology has been observed in a very recent work [115], where a single free fermion chain is monitored through measurements of long-range string operators [116]. Changing the range of the measurement operators, the phase changes from area-law (local measurements) to logarithmic (short-range) to what it seems a regime of mixed logarithmic and volume scaling (long-range). This striking resemblance could be interpreted in terms of the same qualitative features exhibited by long-range measurements and by non-Markovian local measurements – i.e. measurements with a long-time memory that translates into a long spatial range at the level of the system chain. This connection is very interesting and deserves to be accurately investigated on its own, although such a study goes well beyond the scope of this work.

We have also explicitly showed that the effective dynamics of the inner chain is non-Markovian, by computing a measure of non-Markovianity for small systems. The dynamics is always non-Markovian, and the strength of non-Markovianity has a qualitative behavior that resembles that of the entanglement transition. In particular, regions of strong non-Markovianity correspond to regions where the entanglement in the system is larger, suggesting a connection between memory effects and an enhancement of entanglement [77, 117]. We remark that we considered a minimal model of non-interacting bath. Indeed it would be interesting to investigate the role of interactions in the bath, and how they can modify non-Markovianity effects.

## Acknowledgments

We thank M. Fabrizio, R. Fazio, G. Piccitto, J. Piilo, D. Rossini, M. Schiró, E. Tirrito, X. Turkeshi, V. Vitale, S. Kazibwe, and P. Zoller for insightful discussions. G.C. acknowledges the CINECA award under the ISCRA initiative, for the availability of high-performance computing resources and support.

**Funding information**  M. T. thanks the Simons Foundation for supporting his Ph.D. studies through Award 284558FY19 to the ICTP. The work of G. C. and M. D. was partly supported by the ERC under grant number 758329 (AGEnTh), by the MIUR Programme FARE (MEPH), and from QUANTERA DYNAMITE PCI2022-132919. G.C. is supported by ICSC – Centro Nazionale di Ricerca in High-Performance Computing, Big Data and Quantum Computing under project E63C22001000006. M.D. was partly supported by the Munich Institute for Astro-, Particle and BioPhysics (MIAPbP) which is funded by the Deutsche Forschungsgemeinschaft (DFG, German Research Foundation) under Germany´s Excellence Strategy – EXC-2094 – 390783311, and by the PNRR MUR project PE0000023-NQSTI. D.P. acknowledges support from the Ministry of Education Singapore, under the grant MOE-T2EP50120-0019.

## A    Evolution towards the steady state for a single trajectory

In this section, we estimate the time it take for the system to reach the steady state along a single trajectory. After a transient regime of duration $\sim t_{st}$, an observable that evolves along a single trajectory converges to a steady state value around which it fluctuates. The magnitude of these fluctuations depend on the system size and model parameters, but we can typically truncate the evolution of the trajectory at $t = t_{st}$.

The time evolution of a single trajectory entanglement entropy for $l_A = L/2$ bipartition is shown on Fig.(12). The dashed and solid lines correspond to an initial pure state with Néel-

(a) 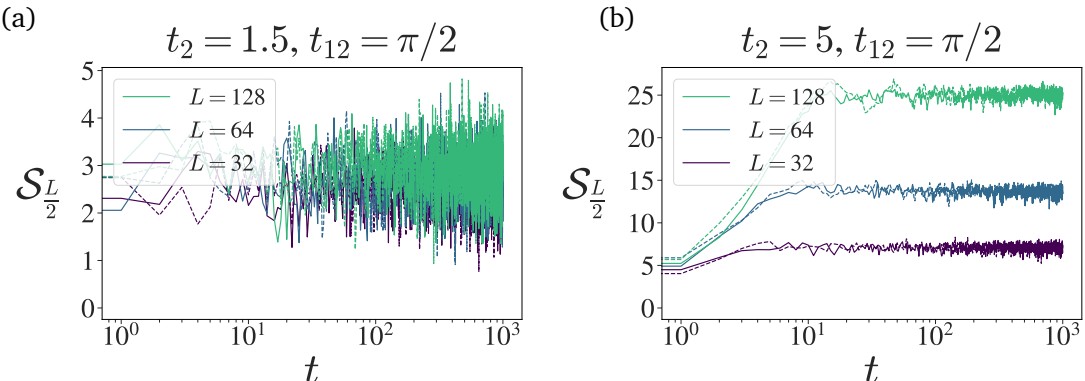 (b)

Figure 12: The time-scaling of entanglement entropy of *A*, with $l_A = L/2$. The solid and dashed lines correspond to random and Neel-like pure state initial conditions.

like (i.e. antiferromagnetic fermionic populations) and random configuration of fermions, respectively. As seen on the figure, the dynamics of both initial conditions yield the same transient and stationary states. For this reason, all of the simulations are performed with random initial conditions, different for every trajectory. The figures show that larger systems take more time to reach the steady state, but for both $t_2 = 1.5$ and $t_2 = 5$, the saturation time for $S_{L/2}$ does not exceed 100 time-steps.

Fig.(13) shows the dynamics of a single trajectory mutual information, between regions *A* and *B* (with $l_A = l_B = L/8$) located opposite to each other, as in Fig.(1). Since the mutual information is a very non-linear function of the density matrix, it is more prone to fluctuations. In the vicinity of the area-law phase, the situation is qualitatively the same as for the entanglement entropy - the mutual information quickly reaches small but non-zero values and rapidly oscillates around it.

The time evolution of a single trajectory mutual information, between regions *A* and *B*, with $l_A = l_B = L/4$, is shown in Fig.(14). The picture is qualitatively the same as in Fig.(13). A difference we observe is that the values of mutual information for both $t_2 = 5$ and $t_2 = 1.5$ are reduced and stronger fluctuations are present. Also, for larger systems, the time it takes for the system to saturate to a steady state seems to exceed 100 time-steps.

Considering these results, we conclude that for $p = 1$, a safe estimate for the amount of time it takes to achieve convergence is $t_{st} = 100$ for small systems ($L \leq 64$) and $t_{st} = 1000$ for larger systems ($L > 64$).

(a) 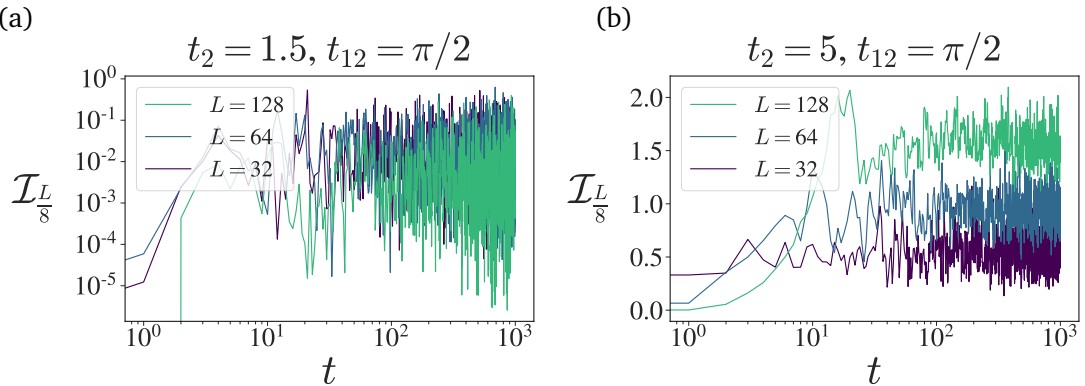 (b)

Figure 13: The time-scaling of mutual information between *A* and *B*, for $l_A = l_B = L/8$ and $r_{AB} = L/2$ distance between them.

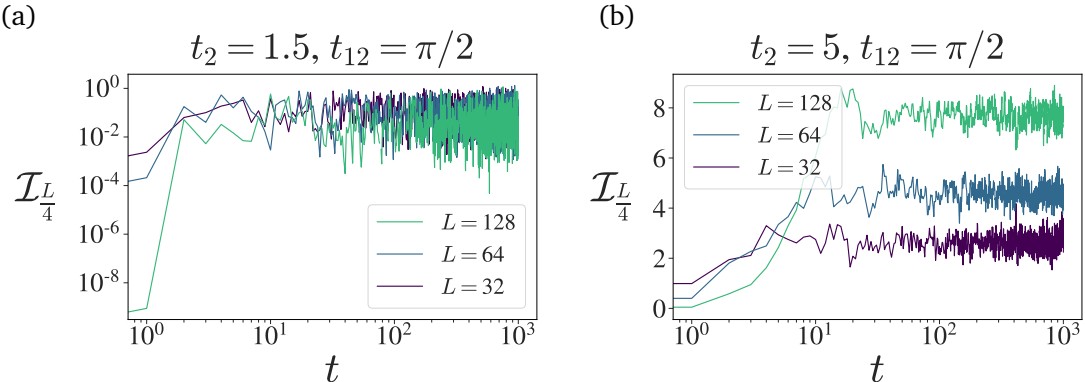

Figure 14: The time-scaling of mutual information between $A$ and $B$, for $l_A = l_B = L/4$ and $r_{AB} = L/2$ distance between them.

In Fig.(15) we present the dynamics of the fermionic negativity for a single trajectory, for $l_A = L/2$ bipartition. The results correspond to a single fixed system size of $L = 128$ and $p = 0.25, 0.5, 0.75$ measurement probabilities. As explained in Sec.(3.2), the peculiar growth of fermionic negativity with respect to $p$ for $t_2 = 5$ is already evident on a single trajectory level, Fig.(15b).

Based on these results, it is safe to fix $t_{st} = 100$ as the time when the system reaches the steady state.

## B The convergence of the ensemble averages with respect to the number of trajectories

In this section we study the convergence of the steady state value of an observables with respect to the number of trajectories $N_{traj}$, with $t_{st} = 100$ for $L \leq 64$ and $t_{st} = 1000$ for $L > 64$. Each trajectory has a different random initial condition.

Fig.(16) shows how $\overline{\mathcal{S}}_{L/2}$ converges with respect to $N_{traj}$. The solid lines represent the average value of the observable, while the shaded regions correspond to a 95% confidence interval calculated from the distribution of $\mathcal{S}_{L/2}$ over the quantum trajectories. The parameters are chosen to be $t_2 = 1.5$ and 5, with fixed $t_{12} = \pi/2$. As it is seen, for a proper convergence, a bigger $N_{traj}$ is needed for larger system sizes and for values of $t_2$ close to the area-law regime $t_2 = 1$.

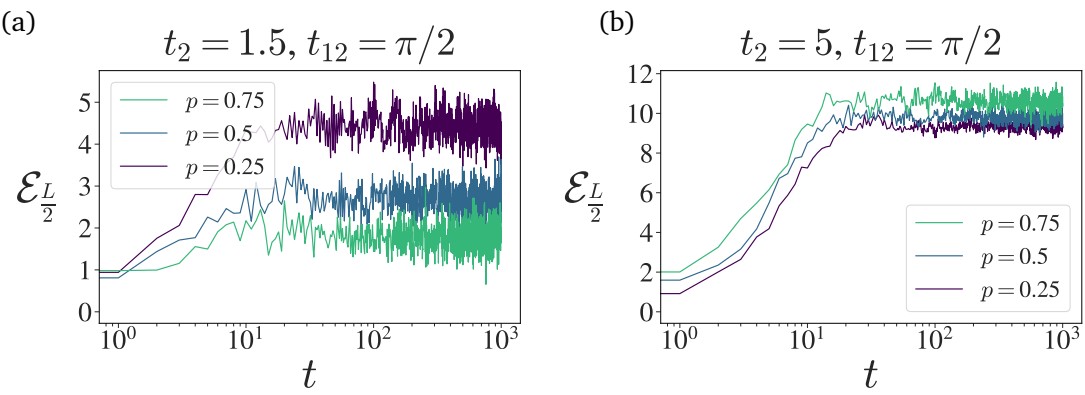

Figure 15: The time-scaling of bipartite negativity of $A$, with $l_A = L/2$.

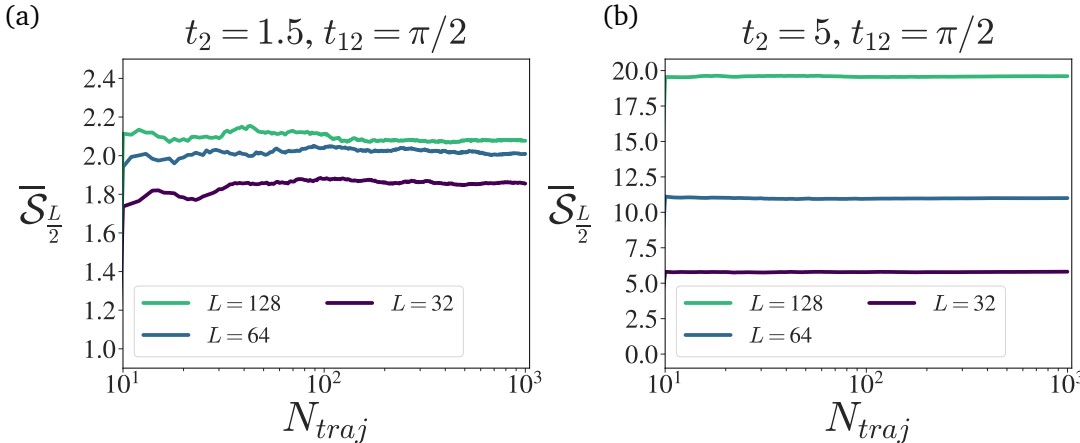

Figure 16: The figures (a) and (b) show the convergence of $\overline{\mathcal{S}}_{L/2}$ with respect to number of trajectories $N_{traj}$.

The convergence of the mutual informations $\overline{\mathcal{I}}_{L/4}$ and $\overline{\mathcal{I}}_{L/8}$ with respect to $N_{traj}$ is illustrated on Figs.(17,18). For $t_2 = 5$, both quantities rapidly saturate to the corresponding average values. However for $t_2 = 1.5$, as Figs.(17a,18a) shows, the mutual informations are more sensitive due to the proximity of an area-law regime and it takes more trajectories for a proper convergence.

Considering these results, we assume that for $p = 1$ and for the regimes far away from the area law-regimes (i.e. for $t_2 > 1.5$), $N_{traj} = 400$ is sufficient for the convergence of the trajectory averaged quantities for $L \leq 64$ and we use $N_{traj} = 1000$ for $L > 64$. For the regions with $t_2 \leq 1.5$, we take $N_{traj} = 1000$ for all system sizes.

Fig.(19) shows how $\overline{\mathcal{E}}_{L/2}$ converges with respect to $N_{traj}$, for various values of measurement probability $p$ and fixed system size $L = 128$. As it is seen, for $p = 0.25, 0.5$ and $0.75$, $N_{traj} \approx 100$ is already sufficient number of trajectories. Thus in order to ensure a proper convergence of our simulations, we set $N_{traj} = 1000$ for $L > 64$ and $N_{traj} = 400$ for $L \leq 64$, regardless of values of $t_2$ and $p$.

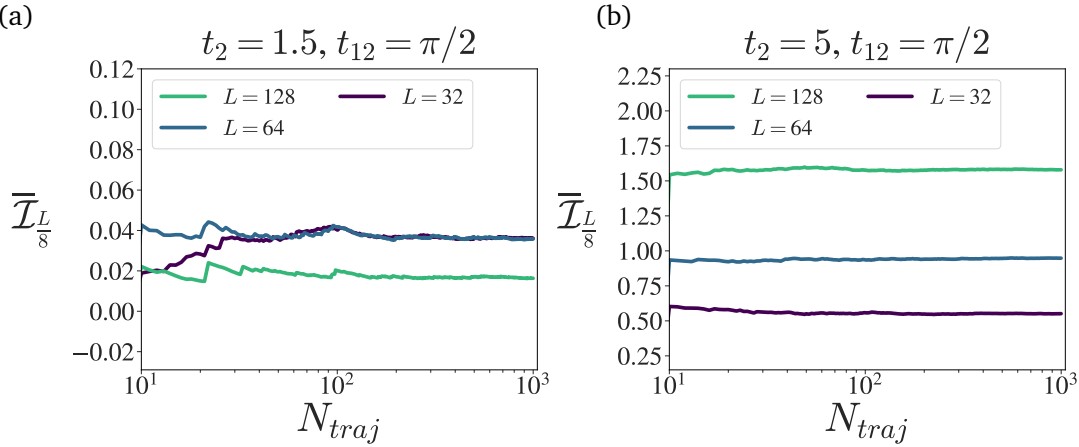

Figure 17: The figures shows the convergence of $\overline{\mathcal{I}}_{L/8}$ with respect to number of trajectories $N_{traj}$.

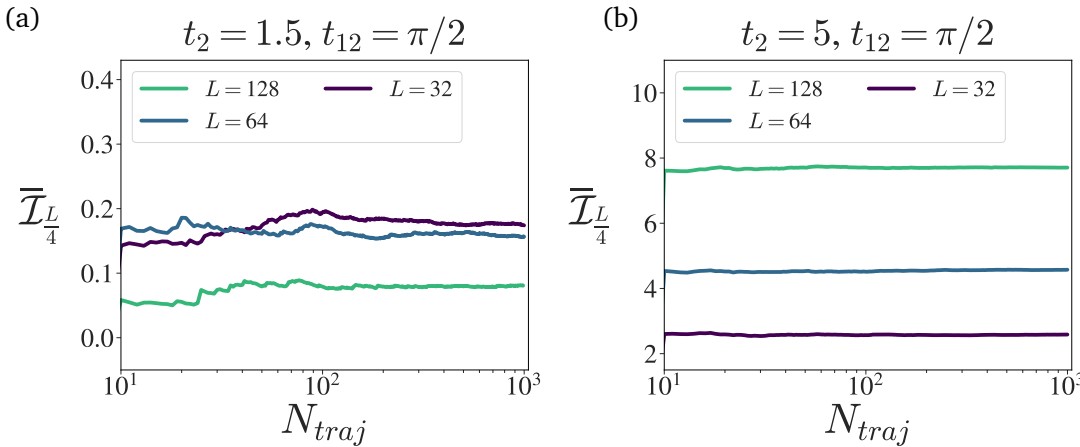

Figure 18: The figures shows the convergence of $\overline{\mathcal{I}}_{L/4}$ with respect to number of trajectories $N_{traj}$.

## C  Finite size effects on the scaling of the entanglement entropy

In this Appendix we show the details of the scaling fit for the entanglement entropy $\overline{\mathcal{S}}_{L/2}$. We fit with the function $\overline{\mathcal{S}}_{L/2} = \gamma L + c/3 \log(L) + \beta$ within a range $L_{\min} \leq L \leq L_{\max}$, for three different intervals $L \in [8, 64]$, $L \in [32, 80]$ and $L \in [72, 128]$. We plot the behavior of $c/3 \log(L_{\max})$ and $\gamma L_{\max}$ as function of $t_2$ for the three different intervals in Fig.(20).

We observe that $\gamma L_{\max}$ is very small below a threshold value $t_2^{\lin}$ and displays a sharp increase after such value. Simultaneously $c/3 \log(L_{\max})$ displays a peak around $t_2^{\lin}$.

We find that the value of $t_2^{\lin}$ increases as $L_{\max}$ is increased, meaning that the region where the linear contribution becomes comparable with the logarithmic contribution is pushed to larger and larger values of $t_2$. Moreover, the maximum value of $\gamma L_{\max}$, observed at larger values of $t_2$, does not change significantly when the fitting interval changes. On the other hand, the peak value of $c/3 \log(L_{\max})$, as well as its value at large $t_2$ increases significantly when $L_{\max}$ is increased. This suggests that in the thermodynamic limit the logarithmic contribution will always dominate, and that the linear contribution is a finite size effect. The crossover size

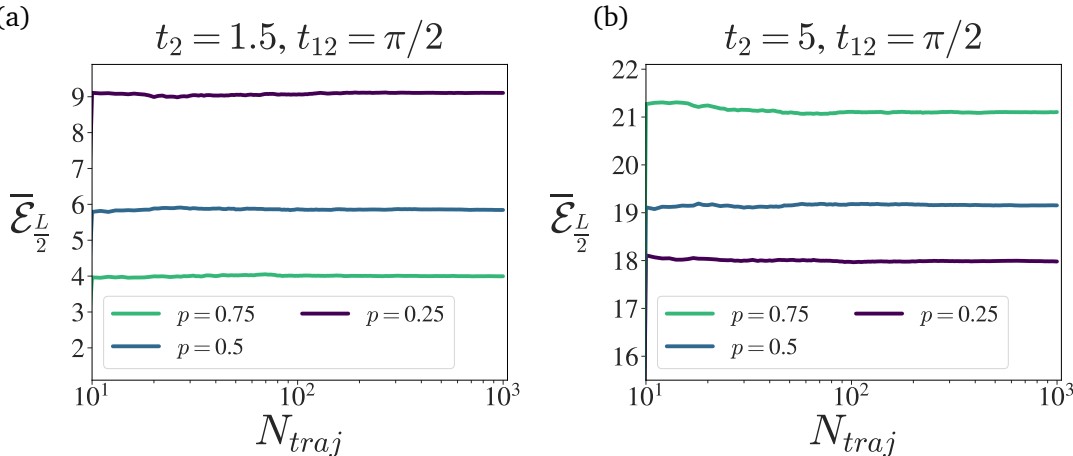

Figure 19: The figures (a) and (b) show the convergence of bipartite logarithmic negativity $\overline{\mathcal{E}}_{L/2}$ with respect to number of trajectories $N_{traj}$, for various $p$ and fixed $L = 128$.

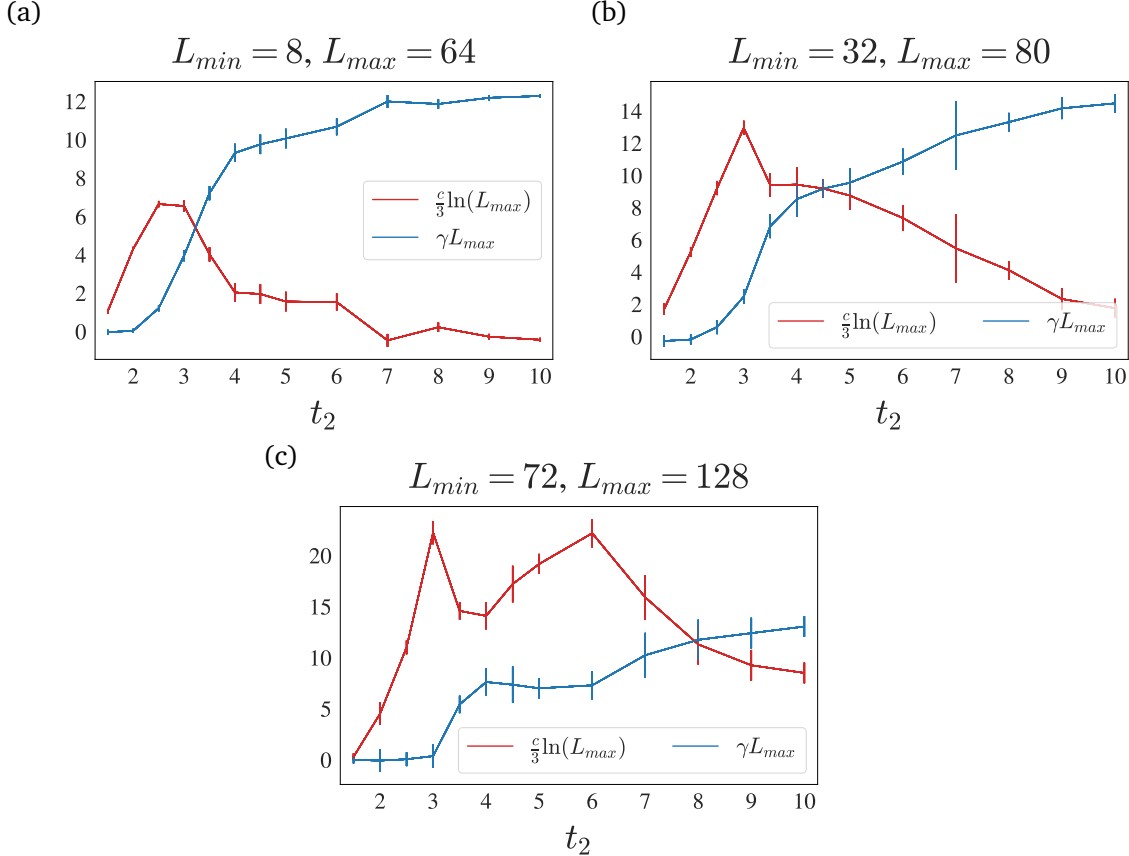

Figure 20: Plot of $c/3\log(L_{\max})$ and $\gamma L_{\max}$ as function of $t_2$ for the three fitting intervals $L \in [8, 64]$ (a), $L \in [32, 80]$ (b) and $L \in [72, 128]$ (c).

$L^*$ above which the logarithmic contribution is dominant seems to be dependent on $t_2$, with larger crossover sizes needed to observe the logarithmic scaling as $t_2$ is increased.

## D Residual analysis of the data fit for negativity

In this section, we analyse the residuals of the data fit for $\overline{\mathcal{E}}_{L/2}$ vs $L$, for fixed $t_{12} = \pi/2$, with various $t_2$ and $p$. We compare the total residual $\Delta\overline{\mathcal{E}}_{L/2} = \sum_{L \in L_{\text{fit}}} \left( \overline{\mathcal{E}}_{L/2}^{(\text{data})} - \overline{\mathcal{E}}_{L/2}^{(\text{fit})} \right)$ for linear and logarithmic fitting curves, for $L$ in fitting range $L_{\text{fit}} = [L_{\min}, L_{\max}]$.

Fig. 21 shows the residuals for linear (dash-dot lines, with $\overline{\mathcal{E}}_{L/2}^{(\text{fit})} = \gamma L + \beta$ fitting curve) and logarithmic (dotted lines, with $\overline{\mathcal{E}}_{L/2}^{(\text{fit})} = c\log(L) + \beta$ fitting curve). As the figures show, for smaller values of $t_2$, the logarithmic fit yields smaller residuals and thus is a better fit, compared to linear. On the other hand, for larger $t_2$, linear fit seems to be more accurate compared to the logarithmic.

It should be noted that the shift of the crossing point between dash-dot and dotted lines by variation of $L_{\text{fit}}$ is attributed to the finite-size effects.

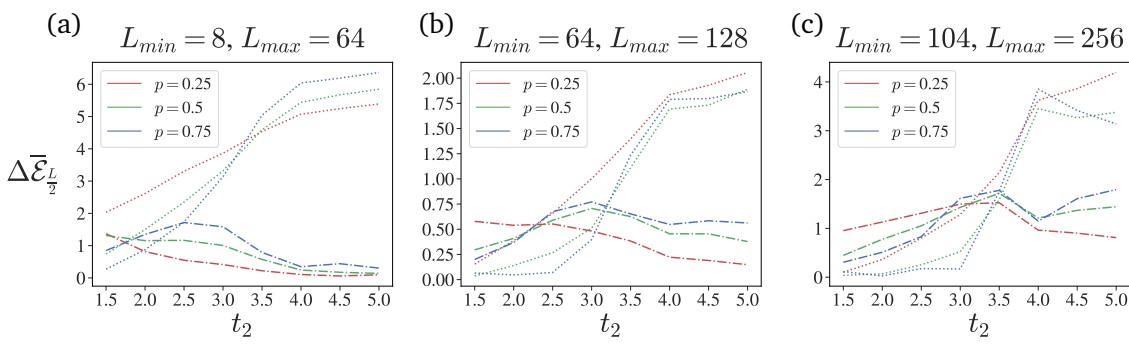

Figure 21

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
