# Peer review of "Measurement induced transitions in non-Markovian free fermion ladders"

_SciPost Physics Core, doi:SciPost Phys. Core 7, 011 (2024)_

## Round 1 · Referee Report · Anonymous (Referee 1) · 2024-2-10

Report

I thank the authors for their reply. I had not appreciated some of the complications in their numerical simulations of this model (e.g. the fact that they are sometimes looking at a statistical mixture of Gaussian states which is not itself Gaussian). Nonetheless the additions and improvements are nice, including the new method based on the Frobenius distance that allows them to push the number of sites from L=4 to L=8 in the analysis of Markovianity. I am in favor of publishing the revised paper.
  • validity: -
  • significance: -
  • originality: -
  • clarity: -
  • formatting: -
  • grammar: -

Author:  Giuliano Chiriacò  on 2024-02-26  [id 4322]

(in reply to Report 1 on 2024-02-10)

We thank the Referee for the positive recommendation on the resubmitted version of our manuscript.

---

## Round 1 · Referee Report · Anonymous (Referee 2) · 2024-2-20

Report

Thank authors for responding to my questions and comments in great details. I am generally OK with the reply and would like to recommend for publication.

However, I realize that the authors might misunderstood my comments on the numerical calculation of $N(\phi)$. Let me clarify myself again here. I agree that sum of Gaussian density matrices, $\rho = \sum_\alpha \rho_\alpha/N_{tr}$, is generally no longer Gaussian. However, since each individual $\rho_\alpha$ is still a Gaussian state, it is not difficult to write $\rho_\alpha$ as a matrix in the Fock space (For example, if $\rho_\alpha = e^{-c^\dagger M c}$, we can first write down the matrix representation of $\rho_\alpha$ in the basis spanned by the single-particle eigenstates of $M$, and then unitarily rotate it to a reference basis). Then one just compute $\rho$ by summing these matrices, and then compute $N(\phi)$ etc. For a chain of length $L$, the dimension of the Fock space is $2^L$, which can be handled easily when $L\leq 12$.

I might miss additional technical difficulties and will be curious of authors' response. If it is indeed a feasible approach, I would strongly recommend the authors to try to extend the numerics to $L=10$ or even $L=12$.
  • validity: -
  • significance: -
  • originality: -
  • clarity: -
  • formatting: -
  • grammar: -

Author:  Giuliano Chiriacò  on 2024-02-26  [id 4323]

(in reply to Report 2 on 2024-02-20)

We thank the Referee for the positive comments on the revised version of our manuscript.

We had actually not understood exactly what the Referee meant in the previous report. We are grateful to the Referee for clarifying it. However, the suggested method of using Gaussian states for the time evolution and then calculating the full density matrix along each trajectory, still contains an exponential complexity like the ED method. This makes it more costly than the method purely based on correlations of Gaussian states, and also more costly than the ED method, which does not require a parallel evolution and then an average over different trajectories. We would also note that the proposed method is feasible for $L\leq12$ in a single chain, which becomes $L\leq6$ in the ladder model we consider, i.e. close to the systems sizes we reach with ED and smaller than the system sizes studied with the purely Gaussian-based method. We have added a brief comment on this in the text, along with a table summarizing the computational complexity of the different methods.

---

## Round 1 · Author Response

Dear Editor,

Thank you for handling our manuscript, and for forwarding the reports provided by the Refer-
ees. We hereby resubmit our manuscript for publication in SciPost Physics Core.
Both Referees had a positive view of the manuscript and think that the investigated system
is interesting and exhibits potential new physics. They had some comments on the technical
side, in particular on the small system sizes of the numerical simulations. We have performed
additional simulations and improved the precision of our numerics.
We hope that with these changes our work is now suitable for publication in SciPost Physics
Core.

Yours sincerely,
M. Tsitsishvili
D. Poletti
M. Dalmonte
G. Chiriacò

---

## Round 1 · List of Changes

1. Modified Fig.(8), now showing the finite size correction to the scaling of the fermionic negativity (page 14).
  2. Also modified the related discussion in the main text (end of page 13 and beginning of page 14).
  3. Added Fig.(11) and corresponding paragraphs to the 4th section ”Measures of non-Markovianity”, where we use the Gaussianity of the state and measure the degree of non-Markovianity using the square trace distance measure (page 17).
  4. Added an appendix D ”Residual Analysis of the data fit for Negativity” (page 25).

---

## Round 2 · Author Response

Dear Editor,

We thank you for handling the manuscript and both the Referees for the positive comments on the resubmitted version of our manuscript.

Regarding the Report of Referee 2, we had actually not understood exactly what the Referee meant in the previous report. We are grateful to the Referee for clarifying it. However, the suggested method of using Gaussian states for the time evolution and then calculating the full density matrix along each trajectory, still contains an exponential complexity like the ED method. This makes it more costly than the method purely based on correlations of Gaussian states, and also more costly than the ED method, which does not require a parallel evolution and then an average over different trajectories. We would also note that the proposed method is feasible for $L\leq12$ in a single chain, which becomes $L\leq6$ in the ladder model we consider, i.e. close to the systems sizes we reach with ED and smaller than the system sizes studied with the purely Gaussian-based method. We have added a remark on this in the text, along with a table summarizing the computational complexity of the different methods.

We hereby resubmit the manuscript with the mentioned change.

Yours sincerely,

M. Tsitsishvili
D. Poletti
M. Dalmonte
G. Chiriacò

---

## Editorial Decision

published